# An Integrative Analysis of Transcriptome, Proteome and Hormones Reveals Key Differentially Expressed Genes and Metabolic Pathways Involved in Flower Development in Loquat

**DOI:** 10.3390/ijms21145107

**Published:** 2020-07-20

**Authors:** Danlong Jing, Weiwei Chen, Ruoqian Hu, Yuchen Zhang, Yan Xia, Shuming Wang, Qiao He, Qigao Guo, Guolu Liang

**Affiliations:** 1Key Laboratory of Horticulture Science for Southern Mountains Regions of Ministry of Education, College of Horticulture and Landscape Architecture, Southwest University, Beibei, Chongqing 400715, China; jingdanlong@swu.edu.cn (D.J.); wenycww@163.com (W.C.); yansummer@swu.edu.cn (Y.X.); wangsm2018@swu.edu.cn (S.W.); swuhq@swu.edu.cn (Q.H.); 2Academy of Agricultural Sciences of Southwest University, State Cultivation Base of Crop Stress Biology for Southern Mountainous Land of Southwest University, Beibei, Chongqing 400715, China; ruoqianhu@126.com (R.H.); zhangyuchen0202@163.com (Y.Z.)

**Keywords:** *Eriobotrya japonica*, transcriptome, flower development, transcription factor, regulatory pathways

## Abstract

Flower development is a vital developmental process in the life cycle of woody perennials, especially fruit trees. Herein, we used transcriptomic, proteomic, and hormone analyses to investigate the key candidate genes/proteins in loquat (*Eriobotrya japonica*) at the stages of flower bud differentiation (FBD), floral bud elongation (FBE), and floral anthesis (FA). Comparative transcriptome analysis showed that differentially expressed genes (DEGs) were mainly enriched in metabolic pathways of hormone signal transduction and starch and sucrose metabolism. Importantly, the DEGs of hormone signal transduction were significantly involved in the signaling pathways of auxin, gibberellins (GAs), cytokinin, ethylene, abscisic acid (ABA), jasmonic acid, and salicylic acid. Meanwhile, key floral integrator genes *FLOWERING LOCUS T* (*FT*) and *SUPPRESSOR OF OVEREXPRESSION OF CONSTANS1* (*SOC1*) and floral meristem identity genes *SQUAMOSA PROMOTER BINDING LIKE* (*SPL*), *LEAFY* (*LFY*), *APETALA1* (*AP1*), and *AP2* were significantly upregulated at the FBD stage. However, key floral organ identity genes *AGAMOUS* (*AG*), *AP3*, and *PISTILLATA* (*PI*) were significantly upregulated at the stages of FBE and FA. Furthermore, transcription factors (TFs) such as *bHLH* (basic helix-loop-helix), *NAC* (no apical meristem (NAM), Arabidopsis transcription activation factor (ATAF1/2) and cup-shaped cotyledon (CUC2)), *MYB_related* (myeloblastosis_related), *ERF* (ethylene response factor), and *C2H2* (*cysteine-2/histidine-2*) were also significantly differentially expressed. Accordingly, comparative proteomic analysis of differentially accumulated proteins (DAPs) and combined enrichment of DEGs and DAPs showed that starch and sucrose metabolism was also significantly enriched. Concentrations of GA_3_ and zeatin were high before the FA stage, but ABA concentration remained high at the FA stage. Our results provide abundant sequence resources for clarifying the underlying mechanisms of the flower development in loquat.

## 1. Introduction

Loquat (*Eriobotrya japonica* Lindl.), an important tropical and subtropical fruit tree species that belongs to the family Rosaceae and is broadly cultivated in many countries such as China, Japan, Spain, the United States, and Australia [1]. Reproductive development of loquat is a continuous process that is not interrupted by winter dormancy [2,3]. Moreover, almost all shoots that develop on the loquat tree are flowering shoots, whereas the number of panicles remains practically constant every year [2]. Thus, it is used as an important material of floral development for evergreen perennials in the Rosaceae family. Therefore, the regulatory mechanisms of floral development in loquat contributes to a better understanding of the flowering process in the Rosaceae family.

Flowering time is critical for reproductive success and is therefore strictly regulated by endogenous and environmental signal pathways [4,5,6,7]. In Arabidopsis, six major pathways, which include photoperiod, vernalization, thermosensory, gibberellin, autonomous, and aging pathways, contribute to the regulation of flower development [8,9,10,11]. For example, the photoperiod and vernalization pathways are involved in light and cold to regulate flowering, respectively, and the flowering response to temperature relies on the thermosensory pathway, which is crucial for mitigating the effects of temperature change [12]. All of these pathways converge to activate a small number of floral integrator genes, which control floral development by merging signals from various pathways [8,13]. Functional analysis of the floral integrators such as *FLOWERING LOCUS T* (*FT*) and *SUPPRESSOR OF OVEREXPRESSION OF CONSTANS 1* (*SOC1*), and floral meristem identity genes *SQUAMOSA PROMOTER BINDING LIKE* (*SPL*), *LEAFY* (*LFY*), and *APETALA1* (*AP1*), have revealed complex regulatory networks of flower development [13,14,15,16,17,18]. Considerable progress in elucidating the flowering mechanism in model plants has greatly improved our understanding of the molecular networks of flower development [9,19,20]. At present, the regulatory mechanisms underlying the flower development of loquat are sorely lacking. 

An integrative analysis of transcriptome and proteome is an extremely effective method for identifying differentially expressed genes (DEGs) from plant developmental phase at the whole genome level [21,22]. This can provide further insight into the underlying mechanisms for flower development in perennial woody plants. At present, transcriptomic analyses have been mainly reported in fruit development in loquat, providing sequence resources that are involved in fruit setting, development, and ripening processes [23,24]. Meanwhile, transcriptomic analysis of flower development has also been used to select the DEGs in some eudicots, such as *Camellia azalea*, *Rosa chinensis*, and *Annona squamosal*. In *C. azalea*, transcriptomic analysis revealed that some floral integrators and MADS-box (MCM1, AGAMOUS, DEFICIENS and SRF) transcription factors (TFs) are involved in floral development [25]. Comparative transcriptome analysis of flower development in *R. chinensis* reported that DEGs are involved in the pathways of circadian clock, sugar metabolism, and hormones [26]. Transcriptomic analysis of *A. squamosal* indicated that various transcription factors are associated with floral transition and flower development [27]. However, an integrative analysis of transcriptome and proteome in the flower development of non-vernalization requiring loquat remains largely unknown and requires further research. 

In this study, we investigated the changes of gene expression dynamics and protein fluctuations during flower development in loquat at three stages, i.e., flower bud differentiation (FBD), flower bud elongation (FBE) and floral anthesis (FA), using transcriptome and proteome sequencing. Analyses of some key DEGs, metabolic pathways, and endogenous hormones highlighted the effect of key flowering genes and metabolic pathways on flower development. Our data provide abundant sequence resources for further clarification of the underlying mechanisms of flower development in loquat and other Rosaceae species. 

## 2. Results

### 2.1. Morphological Characterization of Flower Development

To characterize flower development, we assessed morphological changes during the flower development process. Nine morphologically distinct stages were defined. Vegetative apex development was stage 1, which was embraced by rudimentary leaves (Figure 1S1). Floral meristem initiation and flower bud differentiation were stage 2 (Figure 1S2). Rapid differentiation of floral buds was stage 3, when the top leaves were dark green, and the flower buds elongated longitudinally (Figure 1S3). Stage 4 was rapid panicle elongation (Figure 1S4). Then, floral bud elongation and visible floral buds were stage 5 (Figure 1S5). Stage 6 was the elongation of branches in a panicle (Figure 1S6). Stage 7 was white corollas of floral buds (Figure 1S7). Stage 8 was floral anthesis and full bloom (Figure 1S8). Stage 9 was petal fall (Figure 1S9).

### 2.2. Sequencing, Assembly, and Functional Annotation

To gain insight into the transcriptomic changes during flower development, we prepared cDNA libraries independently from flower bud differentiation (FBD; stage 2), floral bud elongation (FBE; stage 5), and floral anthesis (FA; stage 8). The RNA sequencing of each sample obtained over 6 GB data with Q30 all higher than 92% (Appendix A). A total of 47,234,952 clean reads were obtained for FBD, 47,775,636 for FBE, and 46,826,136 for FA. Assembly of the reads generated 150,219 unigenes, with an average length of 690.3 bp and an N50 of 1164 bp (Appendix A).

A total of 67,072 (44.65%) unigenes were matched to the Nr database by BLAST analysis, while 26,102 (17.38%), 6714 (4.47%), 58,246 (38.77%), and 57,557 (38.32%) were annotated using the Gene Ontology (GO), Kyoto Encyclopedia of Genes and Genomes (KEGG), eggNOG, and Swiss-Prot databases. Among them, 2.49% (3737) of unigenes were assigned to a homolog in all four databases (Appendix A). Meanwhile, a large number of annotated sequences in loquat showed high similarities to the genes in *Malus domestica* and *Prusy* × *bretschneideri*. For example, 29.18% of the unique sequences had top matches to sequences from *M. domestica*, and 23.11% to *P. bretschneideri* (Appendix A).

### 2.3. Analysis of Differentially Expressed Genes (DEGs)

To investigate the DEGs involved in flower development, we analyzed transcript levels of each unigene during three developmental stages (Figure 2). In comparison of FBE vs. FBD, 7409 DEGs were upregulated and 1845 DEGs were downregulated. In the FA vs. FBE comparison, 11,068 differentially expressed, 4406 upregulated, and 6662 downregulated transcripts were detected. In the FA vs. FBD comparison, 13,821 differentially expressed, 8055 upregulated, and 5766 downregulated transcripts were identified. Meanwhile, 730 upregulated and 866 downregulated transcripts were identified in both FBE vs. FBD and FA vs. FBE (Figure 2B,C).

To visualize the DEGs in metabolic pathways, we classified these DEGs on the basis of KEGG pathway enrichment analysis. The DEGs for FBE vs. FBD were mainly enriched in carbon metabolism (37 transcripts), plant hormone signal transduction (33 transcripts), and phenylpropanoid biosynthesis (28 transcripts) (Figure 3A and Appendix A). The DEGs for FA vs. FBE were mainly involved in plant hormone signal transduction (54 transcripts), starch and sucrose metabolism (53 transcripts), and phenylpropanoid biosynthesis (49 transcripts) (Figure 3B and Appendix A). The DEGs for FA vs. FBE were mainly plant hormone signal transduction (72 transcripts), starch and sucrose metabolism (60 transcripts), and phenylpropanoid biosynthesis (48 transcripts) (Appendix A).

### 2.4. Analysis of Key Differentially Expressed Genes (DEGs) Involved in Pathways of Hormone Signal Transduction

Transcriptional levels of hormone signal transduction showed the DEGs involved in the signaling pathways of auxin, gibberellin (GA), cytokinin, ethylene, abscisic acid (ABA), jasmonic acid (JA), and salicylic acid (SA) in three flower development stages in loquat (Figure 4 and Figure 5). During the flower development process, we identified a total of 14 key genes in the auxin-signaling pathway, of which 10 were significantly upregulated at the FA stage (Figure 4A and Table 1). In the GA-signaling and metabolism pathways, we identified seven key genes, of which *GIBBERELLIN 20-OXIDASE 2* (*GA20OX2*), *GIBBERELLIC ACID INSENSITIVE* (*GAI*), and *GIBBERELLIN 3-BETA-DIOXYGENASE 1* (*GA3OX1*) were significantly upregulated at the FBD stage (Figure 4B and Table 1). Meanwhile, four key genes were identified, of which *CYTOKININ OXIDASE/DEHYDROGENASE 5* (*CKX5*), *LONELY GUY 7* (*LOG7*), and *LOG8* were significantly upregulated at the FBE stage (Figure 4C and Table 1).

Compared with FBD, we identified four key genes in the ethylene-signaling pathway, of which *ERF62*, *ERF92*, and *ERF106* were significantly upregulated at the FBE and FA stages (Figure 5A and Table 2). In the ABA-signaling pathway, we identified four key genes, including *ABSCISIC ACID 8’-HYDROXYLASE 2* (*ABAH2*), *ABAH4*, *ABSCISIC ACID-INSENSITIVE 5* (*ABI5*), and *ABSCISIC ACID-INSENSITIVE 5-LIKE PROTEIN 1* (*AI5L1*), of which *ABAH2*, *ABAH4*, and *ABI5* were significantly upregulated at the FBE and FA stages (Figure 5B and Table 2). In the JA-signaling pathway, we identified four key genes, including *JASMONATE O-METHYLTRANSFERASE* (*JMT*), *LIPOXYGENASE 6* (*LOX6*), *LOX15*, and *LOX21*, of which *JMT*, *LOX6*, and *LOX15* were significantly upregulated at the FA stage (Figure 5C and Table 2). In the SA-signaling pathway, we identified *SABP2*, which was significantly upregulated at the FA stage (Figure 5D and Table 2).

### 2.5. Identification of Flowering Pathway-Related Genes and Transcription Factors (TFs)

Expression levels of key genes of flowering pathway and floral molecular networks were further analyzed in flower development. In the vernalization pathway, we identified *VERNALIZATION 1* (*VRN1*), *VERNALIZATION-INDEPENDENT INSENSITIVE 2* (*VIN2*), and *EMBRYONIC FLOWER 2* (*EMF2*), of which *VRN1* and *EMF2* were significantly upregulated at the FBD stage (Figure 6A). In the autonomous pathway, we identified *DICER-LIKE 2* (*DCL2*) and *DCL3*. Meanwhile, in the photoperiod pathway, we identified two unigenes, *LATE ELONGATED HYPOCOTYL* (*LHY*) and *PHYTOCHROME B* (*PHYB*), which were significantly upregulated at the FBD stage (Figure 6A).

The expression levels of eight *SPL*s were significantly upregulated at the FBD stage (Figure 6B). Floral integrator genes *FT* and *SOC1*; floral meristem identity genes *AP1*, *AP2*, and *LFY*; and meristem maintenance gene *WUSCHEL-related homeobox* gene *WOX4* were identified and were found to be significantly upregulated at the FBD stage. Meanwhile, floral organ identity genes *AP3*, *PI*, *AG*, *AGL*, and *SEPALLATA* (*SEP*) were identified and were found to be mainly upregulated at the FBE and FA stages (Figure 6C).

Transcription factors (TFs) are crucial for all stages of flower development. Therein, we identified TF genes regulating flower development in loquat. A total of 8136 TF genes were identified in three development stages (Appendix A). On the basis of the expression levels of TF genes, we further analyzed the five TF families most highly represented in the DEGs. Among them, *bHLH* (basic helix-loop-helix, 874 members), *NAC* (no apical meristem (NAM), Arabidopsis transcription activation factor (ATAF1/2) and cup-shaped cotyledon (CUC2)*,* 624 members), *MYB_related* (myeloblastosis_related, 536 members), *ERF* (ethylene response factor, 508 members), and *C2H2* (cysteine-2/histidine-2, 400 members) genes were identified (Appendix A).

### 2.6. Proteomic Analysis and Identification of Differentially Accumulated Proteins (DAPs)

A total of 32,655 spectra were matched to unique peptides (Appendix A). Ultimately, a total of 8853 proteins were identified in FBD, FBE, and FA stages (Appendix A). On the basis of analysis of GO enrichment, we classified the identified proteins into cellular component, molecular function, and biological process (Figure 7). The biological processes were mainly enriched in the metabolic process (26.87%), cellular process (23.53%), and organic substance metabolic process (20.28%). The molecular functions of proteins were mainly classified into catalytic activity (27.76%) and binding (23.99%). The cellular components were mainly focused on cell (22.46%), cell part (23.02%), and cytoplasm (15.57%). Meanwhile, the identified proteins were mapped to KEGG pathways and were found to be mainly involved in ribosome, protein processing in endoplasmic reticulum, spliceosome, starch and sucrose metabolism, amino sugar and nucleotide sugar metabolism, and RNA transport (Appendix A).

Proteomic analysis of FBE vs. FBD, FA vs. FBE, and FA vs. FBD was used to detect differentially accumulated proteins (DAPs). In the comparison of FBE vs. FBD, we identified 403 DAPs, which were mainly involved in protein processing in the endoplasmic reticulum, photosynthesis, amino sugar and nucleotide sugar metabolism, flavonoid biosynthesis, carbon fixation in photosynthetic organisms, and starch and sucrose metabolism (Appendix A). Comparison of FA vs. FBE showed a total of 684 DAPs were identified and enriched in protein processing in the endoplasmic reticulum, carbon fixation in photosynthetic organisms, phenylpropanoid biosynthesis, starch and sucrose metabolism, and glycolysis/gluconeogenesis (Appendix A). In the FA vs. FBD comparison, we identified a total of 992 DAPs, which were mainly involved in starch and sucrose metabolism, phenylpropanoid biosynthesis, amino sugar and nucleotide sugar metabolism, flavonoid biosynthesis, and glycolysis/gluconeogenesis (Appendix A).

### 2.7. Key Gene Cross-Talk Between the Protein and Transcription Levels

In FBE vs. FBD, FA vs. FBE, and FA vs. FBD, the R values of correlation coefficient were 0.3759, 0.3364, and 0.4138 between proteome and transcriptome (Figure 8A–C), but the R values of correlation coefficient were 0.8514, 0.7742, and 0.8178 between DAPs and DEGs (Figure 8D–F), respectively. Combined analysis of proteome and transcriptome in FBE vs. FBD were mainly enriched in protein processing in the endoplasmic reticulum, photosynthesis, amino sugar and nucleotide sugar metabolism, flavonoid biosynthesis, carbon fixation in photosynthetic organisms, and starch and sucrose metabolism (Figure 9A). The combined enrichment of proteome and transcriptome in FA vs. FBE were mainly involved in protein processing in the endoplasmic reticulum, carbon fixation in photosynthetic organisms, phenylpropanoid biosynthesis, starch and sucrose metabolism, glycolysis/gluconeogenesis, and amino sugar and nucleotide sugar metabolism (Figure 9B). In FA vs. FBD, the combined enrichment of proteome and transcriptome were mainly enriched in starch and sucrose metabolism, phenylpropanoid biosynthesis, amino sugar and nucleotide sugar metabolism, ribosome, flavonoid biosynthesis, and glycolysis/gluconeogenesis (Figure 9C).

### 2.8. Validation of the Expression Levels of Several Key Flower Development-Related Genes

To validate the reliability of the RNA-Seq data, we randomly selected a total of 12 flower development-related unigenes from the DEGs and detected them using qRT-PCR. These DEGs are associated with the flowering-related genes and signaling pathways of plant hormone signal transduction. The same change trends of these DEGs were shown between qRT-PCR and fragments per kilobase per million (FPKM) values (Figure 10), suggesting the expression trends of most unigenes corresponded well between the two methods.

### 2.9. Measurements of Endogenous GA_3_, zeatin (ZT), and ABA Concentrations

GA_3_, ZT, and ABA concentrations were further examined at the FBD, FBE, and FA stages (Figure 11). The GA_3_ and ZT concentrations were high before the FA stage (Figure 11A,B). However, from the FBD to the FBE stage, the ABA concentration increased significantly from 15.67 nmol g^−1^ fresh weight (FW) to 85.40 nmol g^−1^ FW (Figure 11C). This indicated a high correlation between the endogenous hormone concentration changes and expression levels of hormone signal transduction pathway-related genes.

## 3. Discussion

### 3.1. Illumina Sequencing in Flower Development of Loquat

Transcriptomics provides a global analysis of gene fluctuations for analyzing the DEGs during flower development. In this study, we used transcriptome and proteome technology to investigate the changes of key gene expression and proteins during flower development at the FBD, FBE, and FA stages. The results revealed that DEGs include floral integrators, floral meristem identity, floral organ identity genes, and genes/proteins involved in the key pathways of hormone signal transduction and starch and sucrose metabolism associated with flower development (Figure 12). This enabled us to identify key DEGs and metabolic pathways, and further clarify the molecular mechanisms underlying flower development in loquat.

### 3.2. Key Genes of TFs and Floral Integrators Associated with Flower Development

Expression level of floral integrator genes *FT* and *SOC1*; floral meristem identity genes *SPLs*, *LFY*, *AP1*, and *AP2*; and meristem maintenance gene *WOX4* were significantly upregulated at the FBD stage in loquat, suggesting their important roles in the FBD stage. Previous studies have shown that *FT* activates the floral integrator gene *SOC1* and floral meristem gene *AP1* to initiate the floral transition and flower bud differentiation in Arabidopsis [8,28,29]. Then, *SOC1* activates floral meristem identity gene *LFY*, which promotes the phase transition from vegetative to floral meristems [7,8,30]. Meanwhile, *SPLs* directly bind to *SOC1* and *LFY* promoters and activate their transcription and regulate the initiation of floral meristem [31]. Furthermore, *AP2* is regulated by *LFY*, and plays a central role in promoting floral meristem determinacy [32,33,34]. Previously, it was also reported that *FT*, *SOC1*, *SPL4*, *SPL5*, and *SPL9* orthologs are upregulated in flower bud differentiation in loquat [2,35,36]. These similar results indicate that the RNA-Seq data and expression changes of the DEGs involved in floral transition are reliable in our study. On the basis of previously published studies and our data, we also provide a conceptual model for regulatory network of flower development in loquat (Figure 12).

In our work, expression levels of floral organ identity genes *AG*, *AP3*, and *PI* orthologs were mainly upregulated at the FBE stage, suggesting that these genes play important roles in flower organ development in loquat. Previously, *AG*, *AP3*, *PI*, and *AGL* orthologs have been reported to be mainly transcribed in flower organs, playing crucial roles in floral organ identity specification. For example, the expression of *AG* orthologs was strongly detected in reproductive organs including stamens and carpels in different clades of angiosperms, such as *TrAG* of *Taihangia rupestris*, *PsAG* of *Prunus serotine*, and *KjAG* of *Kerria japonica* in rosids [37,38,39]; *TAG1* of *Solanum lycopersicum* and *CpAG* in *Cyclamen persicum* in asterids [40,41]; *MAwuAG* of *Magnolia wufengensis* and *ThtAG1* of *Thalictrum thalictroides* in basal eudicots [42,43]; and *AhMADS6* of *Alpinia hainanensis* and *LMADS10* of lily and in monocots [44,45]. Meanwhile, *AP3* and *PI* orthologs were transcribed only in petals and stamens in other core eudicots, such as *Arabidopsis* [46,47], *Antirrhinum* [48,49], *Torenia fournieri* [50,51], and *S. lycopersicum* [52]. Therefore, we conclude that these presented studies indicate conservative expression pattern of floral organ identity genes between loquat and other eudicots.

### 3.3. Expression Analysis of Key Differentially Expressed Genes (DEGs) Involved in Plant Hormone Signal Transduction Pathways

Plant hormones participate in all stages of flower development. In our study, DEGs involved in auxin-, GA-, cytokinin-, ethylene-, ABA-, JA-, and SA-signaling pathways were significantly enriched during flower development in loquat. Consistent with these results, concentrations of GA_3_ and ZT were high before the FA stage, but ABA concentration remained high at the FA stage. In particular, previous studies have also reported that GA_3_ concentration was significantly changed in homeotic transformation of flower organ in double-flower loquat, and flowers under GA_3_ treatment significantly increased fruit setting in triploid loquat [23,53]. These data indicated that the GA_3_ might play important roles in flower development in loquat.

In plants, GA is involved in various developmental processes, including the initiation of flowering transition, floral primordia, and the identification of floral organs [54,55]. In our work, genes of GA signaling and metabolism pathways and the GA content were mainly upregulated at the FBD stage, indicating the fact that GA plays important roles in the early stage of flower development. In Arabidopsis, GAs control floral induction through regulating floral integrator genes *SOC1* and *FT* and floral meristem identity gene *LFY* [56,57,58]. In *M. domestica*, it was previously reported that the concentration changes of GA are involved in the transition from vegetative buds to floral buds [59]. Previously, transcriptomic analysis of flower development demonstrated that floral organ development is regulated by GA signaling in *Gerbera hybrida* [60]. In *Dendrobium nobile*, comparative transcriptomic analysis revealed that DEGs of GA signaling pathways are involved in floral transition [61]. These previous studies and our data suggest that key genes of GA signaling and metabolism pathways are associated with flower bud differentiation.

Genes of cytokinin signaling pathways and cytokinin content were mainly upregulated at the FBE stage in loquat. In Arabidopsis, cytokinin regulates many aspects of flower growth, and the genes of cytokinin signaling pathways were mainly upregulated in modulating flower development [62,63]. However, cytokinin concentration is high and the genes of cytokinin biosynthesis are upregulated in early flower bud differentiation in *Litchi chinensis* [64,65]. In *Brassica napus*, cytokinin level is significantly increased in vernalization-induced shoot apices and is involved in floral transition [66]. The difference of cytokinin biosynthesis and content between non-vernalization requiring loquat and vernalization requiring *L. chinensis* and *B. napus* might be due to different flower development characteristics or far genetic relationship.

In our study, genes of ABA signaling pathway and ABA content were significantly upregulated after the FA stage. Previously, it was reported that both genes of the ABA signaling pathway and ABA content were upregulated in later stages of flower development in *Lonicera japonica* [67]. In the later stage of flower development, ABA acts as a positive regulator for the senescence [68]. The increase in ABA concentration in the later stage of loquat flower development may be related to the involvement of ABA in petal senescence.

### 3.4. Key DEGs and DAPs Involved in Starch and Sucrose Metabolism Pathways

In our work, comparative transcriptome analysis showed that DEGs were mainly enriched in the pathways of starch and sucrose metabolism. Consistent with the enrichment of DEGs, comparative proteomic analysis and combined enrichment of DEGs and DAPs showed that starch and sucrose metabolism were also significantly enriched in comparisons of FBE vs. FBD, FA vs. FBE, and FA vs. FBD. Similarly, transcription profiles revealed starch metabolism-related genes were upregulated in flower induction under a shoot-bending treatment in *M. domestica* [69]. In *Cucumis sativus*, the DEGs were mainly enriched in the pathway of starch and sucrose metabolism in the induction of female flower [70]. Furthermore, proteomic analysis of fertile and sterile flower buds found that the up- and downregulated proteins were mainly involved in starch and sucrose metabolism in *Brassica campestris* [71]. In *Hemerocallis hybrid*, quantitative proteomic analysis showed DAPs were mainly enriched in starch and sucrose metabolism during flower development [72]. These previous transcriptome and proteomic analysis and our data indicate that starch and sucrose metabolism-related genes play key roles during flower development.

## 4. Materials and Methods

### 4.1. Plant Materials

Flower buds of different development stages were collected from loquat (excellent triploid line ‘CB-1 Q11’) at the experimental farm of Southwest University (Chongqing, China). The loquat trees were 15 years old and were considered to be in the adult phase. Morphological changes of floral bud development were observed from the stages of vegetative meristem to petal fall. Floral buds at three stages, i.e., flower bud differentiation (FBD), floral bud elongation (FBE), and floral anthesis (FA), were collected. At each sampling point, the buds from about 20 panicles were collected and sampled. The bud samples were frozen immediately in liquid nitrogen and stored at −80 °C.

### 4.2. RNA Extraction and Construction of cDNA Library

Total RNA was extracted individually from FBD, FBE, and FA using the EASYspin Plant RNA Extraction kit (RN09, Aidlab, China), according to instructions from the manufacturer. RNA concentration was detected using the NanoDrop 2000 (Thermo Scientific, Wilmington, DE, USA). RNA integrity was detected using the Agilent Bioanalyzer 2100 System (Agilent Technologies, Palo Alto, CA, USA). Then, residual genomic DNA of the RNA was digested using RNase-free DNase I (Takara, Tokyo, Japan) at 37 °C for 30 min. Three independent biological replicates were performed for each developmental stage. The cDNA libraries were constructed using the TruSeqRNA Sample Preparation kit (Illumina, San Diego, CA, USA). Then, the libraries were sequenced using an Illumina HiSeq X-Ten platform (Shanghai Personal Biotechnology Co., Ltd., Shanghai, China).

### 4.3. Data Filtering, de novo Assembly, and Annotation

The sequenced raw reads were processed by removing adapter sequences; ‘N’ bases (unknown bases) and low-quality reads were removed. Then, high-quality clean reads were assembled to contigs, transcripts, and unigenes using the Trinity program [73]. The assembled unigenes were annotated using BLAST with an *E*-value < 0.00001. These BLAST databases included the databases of Swiss-Prot protein, non-redundant protein, Gene Ontology [74], Kyoto Encyclopedia of Genes and Genomes [75], and non-supervised orthologous groups [76].

### 4.4. Identification of Differentially Expressed Genes (DEGs) and Transcription Factors (TFs)

Expression levels of unigenes from three flower developmental stages were compared using the fragments per kilobase per million from the mapped reads (FPKM) by a method described previously [77]. DESeq software was used, and the genes with a |log2 ratio| > 1 and a false discovery rate (FDR) ≤ 0.05 were considered to be significant DEGs [78,79]. Then, DEGs were mapped to the enrichment analysis of GO terms and KEGG pathways using the method described previously [63,64]. GO terms and KEGG pathways showing a corrected *p*-value ≤ 0.05 were considered to be significantly enriched. Searches for TFs were identified using HMMER3.0 (http://hmmer.org/) against the Plant Transcription Factor Database (PlantTFDB, http://planttfdb.cbi.pku.edu.cn/) [80].

### 4.5. Protein Extraction and Digestion

Proteins were extracted from floral buds using the modified method described previously [81]. Flower buds (500 mg) were ground to a fine powder in liquid nitrogen. The powder was suspended in five times volume of trichloroacetic acid/acetone (1:9) and mixed. Then, the mixture was placed at −20 °C for 4 h and centrifuged at 4 °C at 6000× *g* for 40 min, and the supernatant was discarded. Then, the precipitation was washed using pre-cooling acetone and was subsequently air dried. The ≈30 mg powder was added to 30 times volume of SDT buffer (4% SDS, 100 mM Tris-HCl, 1 mM dithiothreitol (DTT); pH 7.6), mixed, and boiled for 5 min. After centrifugation at 14,000× *g* for 40 min at 4 °C, the supernatant was filtered with 0.22 µm filters. The protein content was quantified with the BCA Protein Assay Kit (Bio-Rad, Irvine, CA, USA) and stored at −80 °C.

### 4.6. Protein Digestion and TMT Labeling

Protein samples containing 200 μg of proteins were incorporated into 30 μL SDT buffer. The detergent, DTT, and other low molecular weight components were removed using UA buffer (8 M urea, 150 mM Tris-HCl; pH 8.0) by repeated ultrafiltration (Microcon units, 10 kD). Then, 100 μL iodoacetamide (IAA) (100 mM IAA in UA buffer) was added to block reduced cysteine residues and was incubated for 30 min in darkness. The filters were washed with 100 μL UA buffer three times, and then 100 μL 100 mM triethylammonium bicarbonate (TEAB) buffer twice. Finally, the protein suspensions were digested with 4 μg trypsin (Promega, Madison, WI) in 40 μL triethylammonium bicarbonate (TEAB) buffer overnight at 37 °C, and the resulting peptides were collected as a filtrate. The peptide content was determined by UV light spectral density at 280 nm using an extinction coefficient of 1.1 of 0.1% (g/L) solution on the basis of the frequency of tryptophan and tyrosine in proteins.

Proteins containing 100 μg peptide mixture of each sample was labeled using a tandem mass tag (TMT) Isobaric Label Reagent Set (Thermo Fisher Scientific Inc., Waltham, MA, USA). There were three biological replicates for each sample. A Pierce high pH reversed-phase fractionation kit (Thermo Fisher Scientific Inc., Waltham, MA, USA) was used to fractionate TMT-labeled digest samples into nine fractions by an increasing acetonitrile step-gradient elution.

### 4.7. Low pH nano-LC–MS/MS Analysis

Each fraction was injected for nanoLC–MS/MS analysis. The peptide mixture was loaded onto a reverse phase trap column (Thermo Scientific Acclaim PepMap100, 100 μm × 2 cm, nanoViper C18) connected to the C18-reversed phase analytical column (Thermo Scientific Easy Column, 10 cm long, 75 μm inner diameter, 3 μm resin) in buffer A (0.1% formic acid) and was separated with a linear gradient of buffer B (84% acetonitrile and 0.1% formic acid) at a flow rate of 300 nL/min controlled by IntelliFlow technology.

### 4.8. LC–MS/MS Analysis

LC–MS/MS analysis was performed using a Q Exactive mass spectrometer (Thermo Scientific) coupled to Easy-nLC (Thermo Fisher Scientific Inc.) for 60/90 min. The mass spectrometer was operated in positive ion mode. MS data was acquired using a data-dependent top10 method that dynamically chose the most abundant precursor ions from the survey scan (300–1800 *m/z*) for higher-energy collisional dissociation (HCD) fragmentation. Automatic gain control target was set to 3e6, and maximum inject time was set to 10 ms. Dynamic exclusion duration was 40.0 s. Survey scans were acquired at a resolution of 70,000 at *m/z* 200, and resolution for HCD spectra was set to 17,500 at *m/z* 200, and isolation width was 2 *m/z*. Normalized collision energy was 30 eV. The instrument was run with peptide recognition mode enabled.

### 4.9. Bioinformatics Analysis of Proteomic Data

MS/MS spectra were searched using Mascot software version 2.2 (Matrix Science, London, United Kingdom) embedded into Proteome Discoverer Version 1.4. Proteins were searched against loquat proteome data from the translation of transcriptome data described above. A 1.2-fold cut-off was set to differentially accumulated proteins (DAPs) with a *p*-value < 0.05. The protein sequences of DAPs were blasted against the KEGG database (http://geneontology.org/) to retrieve their KEGG orthologys (KOs) and were subsequently mapped to the pathways. GO enrichment and KEGG pathway enrichment analyses were applied on the basis of Fisher’ exact test. Benjamini–Hochberg correction for multiple testing was further applied to adjust the derived *p*-value. The functional categories and pathways with *p* < 0.05 were considered significant.

### 4.10. Validation Analysis of Transcriptome Data by qRT-PCR

Total RNA was extracted from the flower buds independently at the FBD, FBE, and FA stages. Then, 2 μg of total RNA was used to synthesize the first-strand cDNA using the Primescript RT reagent kit with genomic DNA (gDNA) Eraser (Takara, Japan). The qRT-PCR primers of these selected genes were designed using Oligo 7.0 software and are shown in Appendix A. These primers were synthesized by Sangon Biotech Co., Ltd. (Shanghai, China). qRT-PCR was performed using CFX96 Touch Real Time PCR Detection System (Bio-Rad Laboratories, Hercules, CA, USA) and SYBR Green-based PCR assay. Reaction mixture was cycled as follows: 94 °C for 5 min, followed by 40 cycles of 94 °C for 20 s, 55 °C for 20 s, and 72 °C for 20 s. Three biological replicates were conducted for each sample. The *actin* gene of loquat was used as a normalization control [82]. The relative quantification of the tested genes was determined by the comparative C_t_ values and calculated by 2^−∆∆Ct^ values [83].

### 4.11. Determination of GA_3_, ZT, and ABA Concentrations

The GA_3_, ZT, and ABA concentrations from FBD, FBE, and FA were determined through using the method described by [53,63,84]. Then, 0.3 g of each sample was extracted. Standard preparations were obtained from gibberellic acid, zeatin, and abscisic acid (Sigma Chemical Co., St. Louis, MO, USA). Then, the concentrations of GA_3_, ZT, and ABA were detected using an Agilent 1260 HPLC device (Agilent Technologies, Santa Clara, CA, USA) equipped with a G1314B UV detector. Three biological replicates were performed. All data were analyzed using analysis of variance (ANOVA), and the differences were compared using PASW Statistics v18.0 software (SPSS Inc., Chicago, IL, USA) and Duncan’s multiple range test.

## 5. Conclusions

Different from previous transcriptome analysis of loquat fruit development [23,24], our results provide key DEGs that were found to be mainly involved in flowering-related genes and the pathways of plant hormone signal transduction and starch and sucrose metabolism during flower development in loquat. Among them, key floral integrator genes and floral meristem identity genes were significantly upregulated at the FBD stage, and floral organ identity genes were significantly upregulated at the stages of FBE and FA. The DEGs were significantly enriched in the signaling pathways of auxin, GA, cytokinin, ethylene, ABA, JA, and SA. Consistent with these results, comparative proteomic analysis of differentially accumulated proteins (DAPs) and combined enrichment of DEGs and DAPs showed that starch and sucrose metabolism were also significantly enriched. Concentrations of GA_3_ and ZT were high before the FA stage, but ABA concentration remained high at the FA stage. Future work should be performed to investigate the possible roles of key DEGs and metabolic pathways in flower development in loquat. Taken together, these identified key genes, hormone signal transduction, and starch and sucrose metabolism pathways increase our understanding of the complex regulatory networks underlying flower development in loquat and other Rosaceae species.

## Figures and Tables

**Figure 1 ijms-21-05107-f001:**
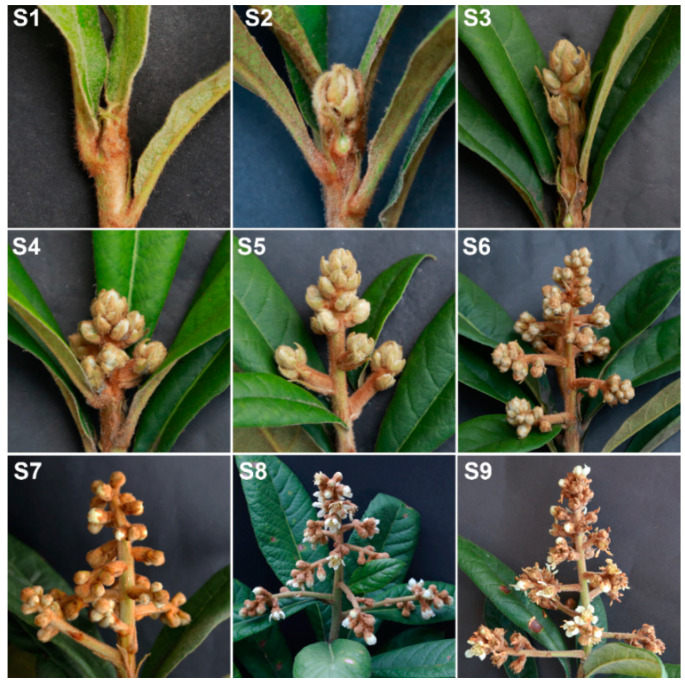
Morphological changes of flower development in loquat (excellent triploid line ‘CB-1 Q11’). (**S1**) Vegetative apex. (**S2**) Floral meristem initiation and flower bud differentiation. (**S3**) Rapid differentiation of floral buds. (**S4**) Panicle elongation. (**S5**) Floral bud elongation with visible floral buds. (**S6**) Elongation of branches in a panicle. (**S7**) White corollas of floral buds. (**S8**) Floral anthesis and full bloom. (**S9**) Petal fall.

**Figure 2 ijms-21-05107-f002:**
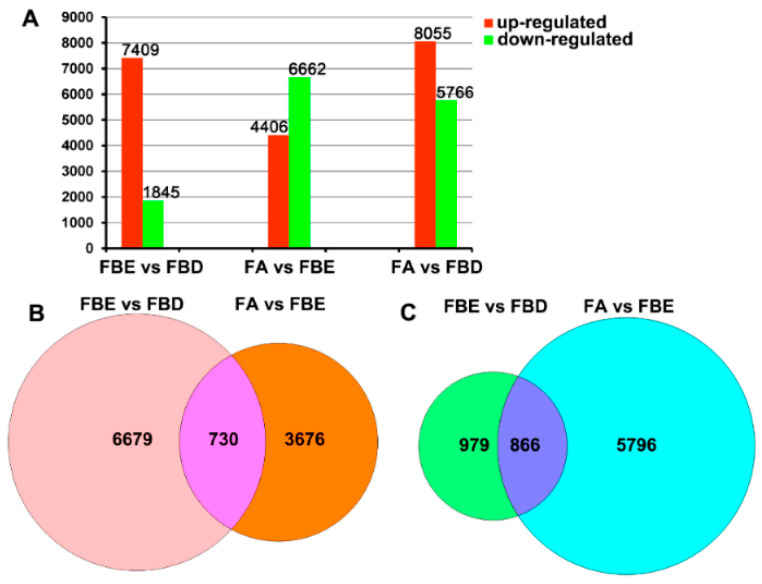
Numbers of differentially expressed genes (DEGs) involved in phase changes during flower development of flower bud differentiation (FBD), floral bud elongation (FBE) and floral anthesis (FA). (**A**) The numbers of DEGs in FBE vs. FBD, FA vs. FBE and FA vs. FBD. (**B**) The numbers of upregulated DEGs in FBE vs. FBD and FA vs. FBE. (**C**) The numbers of downregulated DEGs in FBE vs. FBD and FA vs. FBE.

**Figure 3 ijms-21-05107-f003:**
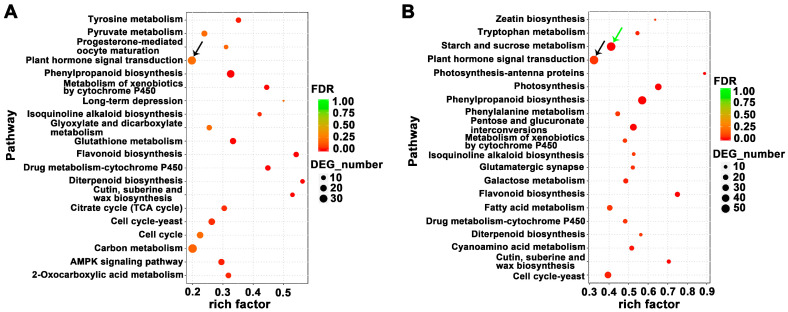
Kyoto Encyclopedia of Genes and Genomes (KEGG) pathway enrichment analysis of DEGs. (**A**) Enrichment analysis of DEGs for floral bud elongation (FBE) vs. flower bud differentiation (FBD). The pathway of plant hormone signal transduction was mainly enriched (black arrow). (**B**) Enrichment analysis of DEGs for floral anthesis (FA) vs. FBE. The pathways of plant hormone signal transduction (black arrow) and starch and sucrose metabolism (green arrow) were mainly enriched. Rich factor is a ratio of the number of DEGs annotated with a pathway relative to the total number of genes annotated with this pathway. The larger value of the rich factor represented, the greater the enrichment of this KEGG pathway.

**Figure 4 ijms-21-05107-f004:**
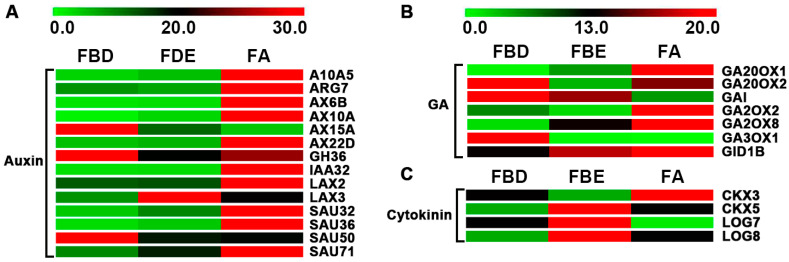
Expression level changes of the DEGs involved in the auxin, gibberellin (GA), and cytokinin signaling and metabolism pathways in three flower development stages in loquat. (**A**) Expression levels of DEGs of auxin signaling pathways. (**B**) Expression levels of DEGs of GA signaling and metabolism pathways. (**C**) Expression levels of DEGs of cytokinin signaling pathways.

**Figure 5 ijms-21-05107-f005:**
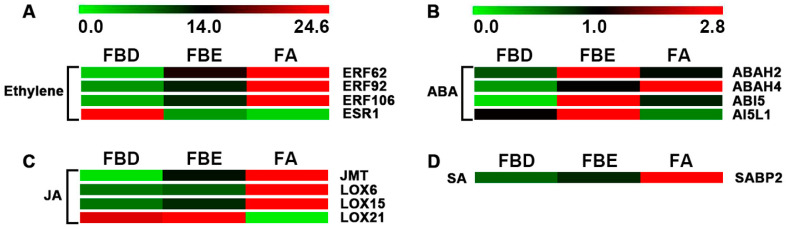
Expression level changes of the DEGs involved in the ethylene-, abscisic acid (ABA)-, jasmonic acid (JA)-, and salicylic acid (SA)-signaling pathways. (**A**) Expression levels of DEGs of ethylene signaling pathways. (**B**) Expression levels of DEGs of ABA signaling pathways. (**C**) Expression levels of DEGs of JA signaling pathways. (**D**) Expression levels of DEGs of SA signaling pathways.

**Figure 6 ijms-21-05107-f006:**
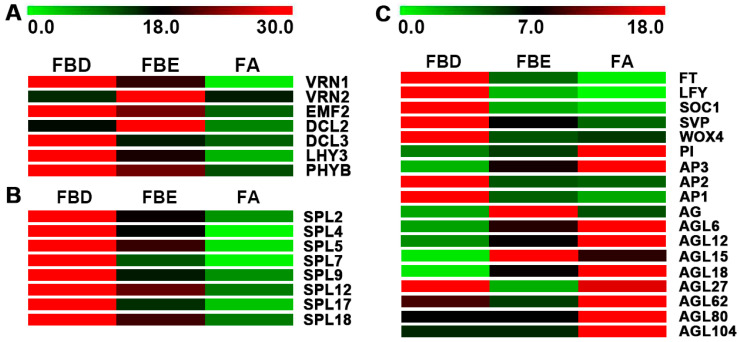
Expression changes of the genes involved in flowering-related pathways, floral integrator, floral meristem identity, and floral organ identity in flower development. (**A**) Expression levels of DEGs involved in flowering-related pathways. (**B**) Expression levels of DEGs encoding SPLs. (**C**) Expression levels of DEGs involved in floral integrator, floral meristem identity, and floral organ identity.

**Figure 7 ijms-21-05107-f007:**
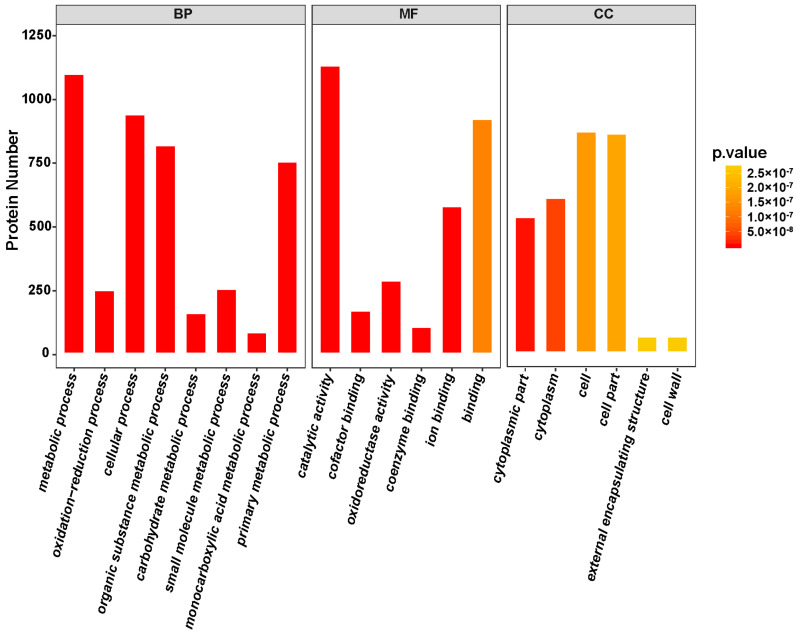
Gene Ontology (GO) enrichment analysis of the identified proteins.

**Figure 8 ijms-21-05107-f008:**
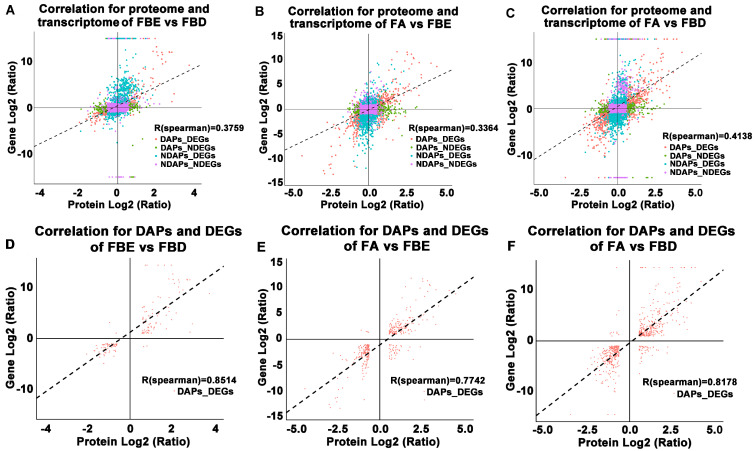
Correlation for proteome and transcriptome, differentially accumulated proteins (DAPs), and DEGs of FBE vs. FBD, FA vs. FBE, and FA vs. FBD. (**A**–**C**) Correlation for proteome and transcriptome of FBE vs. FBD, FA vs. FBE, and FA vs. FBD. (**D**–**F**) Correlation for DAPs and DEGs of FBE vs. FBD, FA vs. FBE, and FA vs. FBD.

**Figure 9 ijms-21-05107-f009:**
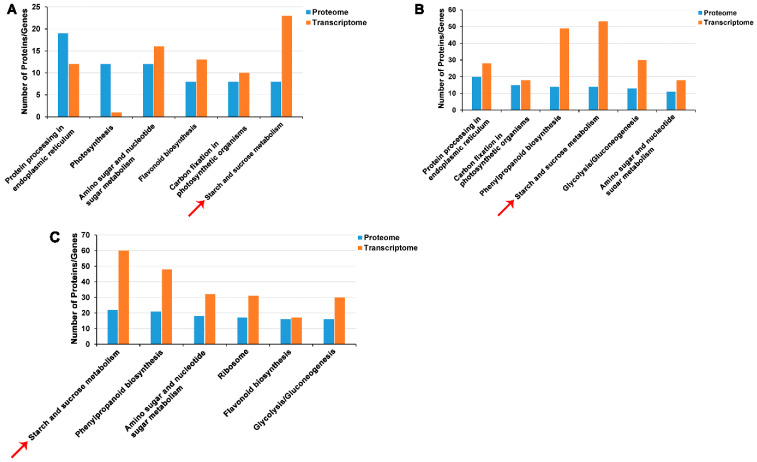
Combined enrichment of proteome and transcriptome in FBE vs. FBD, FA vs. FBE, and FA vs. FBD. (**A**) Combined enrichment of proteome and transcriptome in FBE vs. FBD. (**B**) Combined enrichment of proteome and transcriptome in FA vs. FBE. (**C**) Combined enrichment of proteome and transcriptome in FA vs. FBD. Starch and sucrose metabolism is marked by red arrows.

**Figure 10 ijms-21-05107-f010:**
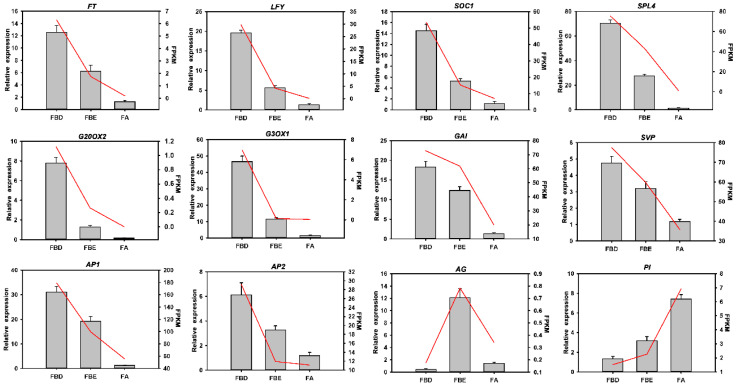
Validation of the expression of flower development-related genes by qRT-PCR analysis. Bar charts indicate values of qRT-PCR. Line plots indicate values of fragments per kilobase per million (FPKM). Error bars indicate the standard deviation of three biological replicates.

**Figure 11 ijms-21-05107-f011:**
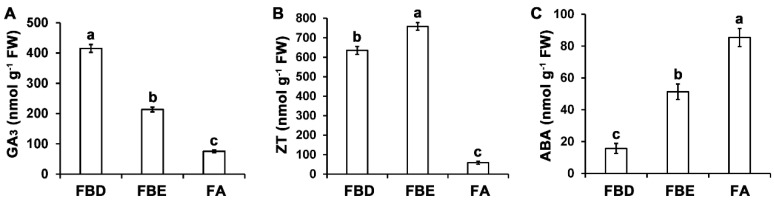
Changes in GA_3_, zeatin (ZT), and ABA concentrations during flower development. (**A**) GA_3_ concentration. (**B**) ZT concentration. (**C**) ABA concentration. Significant differences are indicated by different letters (*p* < 0.05).

**Figure 12 ijms-21-05107-f012:**
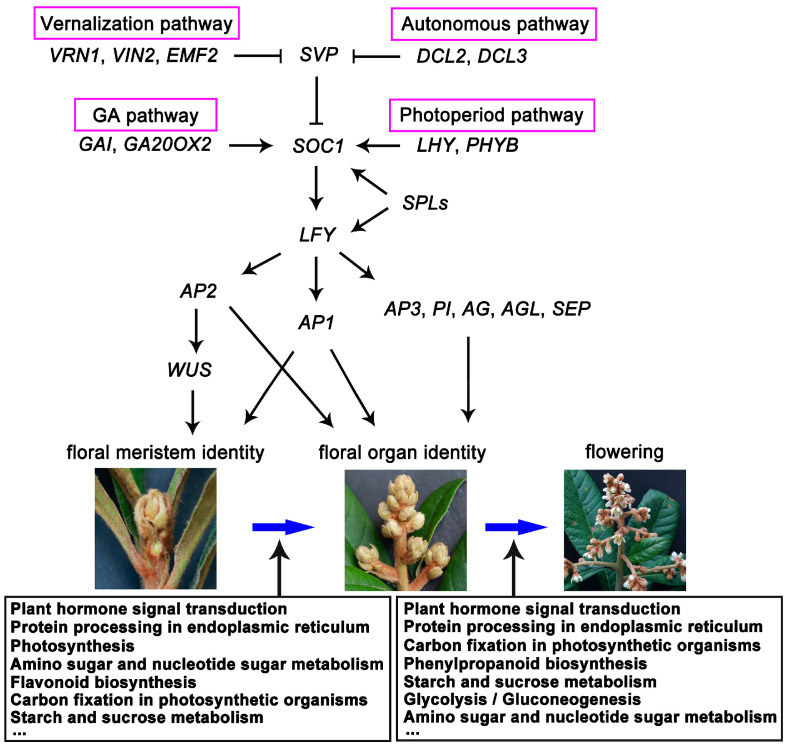
Schematic of the regulatory genes and metabolic pathways in flower development in loquat.

**Table 1 ijms-21-05107-t001:** Differentially expressed genes (DEGs) involved in the signaling pathways of auxin, GA, and cytokinin.

Gene_ID	Annotation	Fold Change (FBE/FBD)	Fold Change (FA/FBE)	Fold Change (FA/FBD)
**Auxin signaling pathway**				
DN62407_c0_g2	Auxin-induced protein 10A5 (A10A5)	1.50	4.88	7.35
DN53030_c0_g1	Indole-3-acetic acid-induced protein ARG7 (ARG7)	0.83	3.35	2.77
DN68900_c2_g4	Auxin-induced protein 6B (AX6B)	1.23	9.02	11.11
DN68900_c3_g2	Auxin-induced protein X10A (AX10A)	3.43	6.90	23.65
DN63607_c3_g1	Auxin-induced protein 15A (AX15A)	0.14	0.39	0.05
DN65976_c2_g3	Auxin-induced protein 22D (AX22D)	1.21	4.76	5.74
DN72894_c3_g5	Indole-3-acetic acid-amido synthetase GH3.6 (GH36)	0.51	1.49	0.76
DN39057_c0_g1	Auxin-responsive protein IAA32 (IAA32)	1.13	7.82	8.82
DN63908_c0_g1	Auxin transporter-like protein 2 (LAX2)	1.05	2.29	2.40
DN62802_c0_g1	Auxin transporter-like protein 3 (LAX3)	2.77	0.85	2.37
DN70362_c2_g3	Auxin-responsive protein SAUR32 (SAU32)	2.25	2.44	5.49
DN70025_c0_g2	Auxin-responsive protein SAUR36 (SAU36)	1.58	5.52	8.74
DN70819_c6_g3	Auxin-responsive protein SAUR50 (SAU50)	0.42	1.07	0.45
DN66182_c1_g5	Auxin-responsive protein SAUR71 (SAU71)	1.85	1.42	2.64
**GA signaling and metabolism pathway**				
DN62963_c0_g1	Gibberellin 20 oxidase 1 (GA20OX1)	13.47	2.67	35.97
DN69182_c1_g4	Gibberellin 20 oxidase 2 (GA20OX2)	0.30	0.28	0.08
DN70029_c1_g4	Gibberellic acid-insensitive (GAI)	0.85	0.32	0.27
DN53975_c0_g1	Gibberellin 2-beta-dioxygenase 2 (GA2OX2)	3.97	10.70	42.43
DN66786_c1_g1	Gibberellin 2-beta-dioxygenase 8 (GA2OX8)	9.42	3.48	32.78
DN52581_c0_g1	Gibberellin 3-beta-dioxygenase 1 (GA3OX1)	0.02	0.27	0.01
DN65482_c3_g1	Gibberellin receptor GID1B ( GID1B)	1.93	1.30	2.50
**Cytokinin signaling pathway**				
DN69821_c1_g5	Cytokinin oxidase/dehydrogenase 3 (CKX3)	0.32	11.69	3.76
DN58587_c0_g1	Cytokinin oxidase/dehydrogenase 5 (CKX5)	29.33	0.10	2.97
DN64500_c1_g3	Lonely guy 7 (LOG7)	2.34	0.03	0.08
DN58587_c0_g1	Lonely guy 8 (LOG8)	29.33	0.10	2.97

**Table 2 ijms-21-05107-t002:** DEGs involved in the ethylene-, ABA-, JA-, and SA-signaling pathways.

Gene_ID	Annotation	Fold Change (FBE/FBD)	Fold Change (FA/FBD)	Fold Change (FA/FBE)
**Ethylene signaling pathway**				
DN3184_c0_g1	Ethylene-responsive transcription factor ERF62 (ERF62)	5.81	13.97	2.40
DN65937_c1_g5	Ethylene-responsive transcription factor ERF92 (ERF92)	2.07	3.60	1.74
DN65178_c3_g5	Ethylene-responsive transcription factor ERF106 (ERF106)	2.60	9.34	3.60
DN66697_c3_g4	Ethylene-responsive transcription factor ESR1 (ESR1)	0.19	0.09	0.46
**ABA signaling pathway**				
DN49553_c0_g1	Abscisic acid 8’-hydroxylase 2 (ABAH2)	2.65	1.38	0.52
DN66767_c1_g1	Abscisic acid 8’-hydroxylase 4 (ABAH4)	3.12	19.82	6.36
DN67864_c2_g4	Abscisic acid-insensitive 5 (ABI5)	13.63	6.27	0.46
DN48609_c0_g1	Abscisic acid-insensitive 5-like protein 1 (AI5L1)	2.66	0.45	0.17
**JA signaling pathway**				
DN68639_c0_g1	Jasmonate O-methyltransferase (JMT)	7.70	61.13	7.94
DN65561_c1_g2	Lipoxygenase 6 (LOX6)	1.20	3.33	2.78
DN68565_c0_g4	Lipoxygenase 15 (LOX15)	1.50	7.72	5.14
DN71255_c1_g1	Lipoxygenase 21 (LOX21)	1.10	0.05	0.05
**SA signaling pathway**				
DN62662_c1_g1	Salicylic acid-binding protein 2 (SABP2)	1.37	2.33	1.70

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
