# Peer review of "An Integrative Analysis of Transcriptome, Proteome and Hormones Reveals Key Differentially Expressed Genes and Metabolic Pathways Involved in Flower Development in Loquat"

_ijms, 2020, doi:10.3390/ijms21145107_

Round 1
Reviewer 1 Report
I read the manuscript "An Integrative Analysis of Transcriptome, Proteome
and Hormone Reveals Key Differentially Expressed
Genes and Metabolic Pathways Involved in Flower
Development in Loquat" by Jing et al. In this work the authors conduct a transcriptome and proteome analysis of loquat. Generally I think the work is valuable, however I think the results should be presented in greater depth - at the moment the observations are rather trivial.
Specifically:
2.1. I am unsure whether the different stages are stages are deducted from previous studies. If not, the authors should properly define each of the stage, and explain their classification system (which seems entirely morphological/visual at the moment
2.2. I think it is trivial to denote that the identified genes matched with other plant species when doing tree transcriptomics. What would be non plant or novel genes - how complete is the transcriptome versus other trees or Arabidopsis?
Figure 2: Abbreviations should be explained in the caption. Applies to all figures
Figure 3: Unclear what rich means
2.4. It is unclear what would be the neutral expectation of differentially expressed genes given the data. (how many genes do you expect to be differentially expressed in these specific pathways)
Figure 4 and following: Visualise the pathways. Where are the pathways derived from (e.g. in what plant do they occur - Arabidopsis?)
2.5 "Based on the expression levels of TF genes, we
further analyzed the five TF families most highly represented in the DEGs" - where is this analysis?
2.6 does is matter what software you use? Mascot software?
2.9. do these genes relate to any of the above mentioned pathways?
check ref 61
Author Response
Dear Editor and Reviewers,
First of all, thank you very much for the minor revisions of your letter and the reviewers’ advice and comments to improve the manuscript “An Integrative Analysis of Transcriptome, Proteome and Hormone Reveals Key Differentially Expressed Genes and Metabolic Pathways Involved in Flower Development in Loquat” (Manuscript ID: ijms-848521-Minor Revisions). These comments are all valuable and very helpful for improving our paper, as well as the important guiding significance to our study. We have studied the comments carefully, and revised our manuscript according to your comments. We hope our revised manuscript has met with the approval. The edited comments were highlighted using red text in the revised manuscript. The main corrections in the manuscript and the responds to your comments are as following.
Now, we give a point-by-point response to the comments:
Response to Reviewer 1
Point 1: I read the manuscript "An Integrative Analysis of Transcriptome, Proteome and Hormone Reveals Key Differentially Expressed Genes and Metabolic Pathways Involved in Flower Development in Loquat" by Jing et al. In this work the authors conduct a transcriptome and proteome analysis of loquat. Generally I think the work is valuable, however I think the results should be presented in greater depth - at the moment the observations are rather trivial.
Response 1: Thanks very much for your positive affirmation and evaluation for our study. We have revised manuscript as your good suggestions. Meanwhile, we have revised the inaccurate English expression and added the data analysis to improve the manuscript. For example, we have changed “has been also” to “has also been”, “floral bud” to “floral buds” and “shown” to “showed” in our revised manuscript. We have also added the correlation analysis for DAPs and DEGs of FBE vs FBD, FA vs FBE and FA vs FBD in Fig. 8. Now we give a point-by-point response to the comments below.
Point 2: 2.1. I am unsure whether the different stages are stages are deducted from previous studies. If not, the authors should properly define each of the stage, and explain their classification system which seems entirely morphological/visual at the moment.
Response 2: In our previous study, we only described the anatomy of the different stages in Jing, et al. 2020. [“Jing D., Chen W., Shi M., Wang D., Xia Y., He Q., Dang J., Guo Q., Liang G. (2020). Ectopic expression of an Eriobotrya japonica APETALA3 ortholog rescues the petal and stamen identities in Arabidopsis ap3-3 mutant. Biochemical and biophysical research communications, 523(1): 33–38.”]. However, different from our previous study, morphological classification system of this study mainly focused on describing visual characteristics and is supplemented for our previous study.
Point 3: 2.2. I think it is trivial to denote that the identified genes matched with other plant species when doing tree transcriptomics. What would be non plant or novel genes-how complete is the transcriptome versus other trees or Arabidopsis?
Response 3: Thanks for this very good suggestion. According to the advice, we have revised the inaccurate English expression and changed “A large number of annotated sequences in loquat showed high similarities to the genes in other plant species” to “a large number of annotated sequences in loquat showed high similarities to the genes in Malus domestica and Prusy × bretschneideri” in our revised manuscript.
Point 4: Figure 2: Abbreviations should be explained in the caption. Applies to all figures
Response 4: Thanks for this very good suggestion. According to the advice, we have added the abbreviations in the caption and applied to all figures. For example, we have spell out DEGs “differentially expressed genes” in L124, L139, L151, L291 to be helpful to the reader.
Point 5: Figure 3: Unclear what rich means.
Response 5: In our revised manuscript, we have changed “rich” to “rich factor”. Meanwhile, we have added the caption “Rich factor is a ratio of the number of DEGs annotated with a pathway relative to the total number of genes annotated with this pathway. The larger value of the rich factor are represented the greater the enrichment of this KEGG pathway” in Figure 3.
Point 6: 2.4. It is unclear what would be the neutral expectation of differentially expressed genes given the data. (how many genes do you expect to be differentially expressed in these specific pathways).
Response 6: In our study, there were the average of five differentially expressed genes in these specific pathways.
Point 7: Figure 4 and following: Visualise the pathways. Where are the pathways derived from (e.g. in what plant do they occur - Arabidopsis?)
Response 7: According to the advice, these DEGs were classified using BLAST with the KEGG databases. We have changed to “Expression level changes of the DEGs involved in the auxin, GA and cytokinin signaling pathways in three flower development stages in loquat”.
Point 8: 2.5 "Based on the expression levels of TF genes, we further analyzed the five TF families most highly represented in the DEGs" - where is this analysis?
Response 8: Thanks for this very good suggestion. According to the advice, we have added the analysis of TF families “Searches for TFs were identified using HMMER3.0 (http://hmmer.org/) against the Plant Transcription Factor Database (PlantTFDB, http://planttfdb.cbi.pku.edu.cn/) [78].” in the methods in our revised manuscript. Meanwhile, we have added a references involved in this analysis: Jin, J.P.; Zhang, H.; Kong, L.; Gao, G.; Luo, J.C., PlantTFDB 3.0: a portal for the functional and evolutionary study of plant transcription factors. Nucleic Acids Research 2014, 42, (D1), D1182-D1187.
Point 9: 2.6 does is matter what software you use? Mascot software?
Response 9: According to the advice, we have revised the inaccurate English expression and deleted “According to Mascot software” in our revised manuscript.
Point 10: 2.9. do these genes relate to any of the above mentioned pathways?
Response 10: In our study, proteins were searched against proteome data from the translation of transcriptome data, therefore the proteins and genes ID number. We have added the “Proteins were searched against loquat proteome data from the translation of transcriptome data described above” in the methods and added the correlation coefficients between DAPs and DEGs, and changed to “In FBE vs FBD, FA vs FBE and FA vs FBD, the R values of correlation coefficient were 0.3759, 0.3364 and 0.4138 between proteome and transcriptome (Figure 8A-C), but the R values of correlation coefficient were 0.8514, 0.7742 and 0.8178 between DAPs and DEGs (Figure 8D-F)”.
Meanwhile, the identified DEGs/DAPs were mapped to KEGG pathways and they were both enriched in metabolic pathways of starch and sucrose metabolism in Figure 3, S2 and 9.
Point 11: check ref 61
Response 11: Thanks for this very good suggestion. We have checked the reference, and changed to “Bartrina, I.; Otto, E.; Strnad, M.; Werner, T.; Schmülling, T., Cytokinin regulates the activity of reproductive meristems, flower organ size, ovule formation, and thus seed yield in Arabidopsis thaliana. The Plant Cell 2011, 23, (1), 69-80”.
We try our best to improve the manuscript and look forward to your positive response. Thank you very much for your consideration.
Danlong Jing
Postal address: College of Horticulture and Landscape Architecture, Southwest University, Beibei, Chongqing 400715, China.
Email: jingdanlong110@126.com
Tel.: +86 023 68250383

Reviewer 2 Report
This is a high quality manuscript based upon a comprehensive, well designed study that presents many lines of converging evidence. The work provides considerable data that enlightens the molecular/genetic knowledge base supporting floral initiation and development in Loquat. It very nicely links gene action to hormonal changes and morphological development in early flowering. The paper is well-written with reasonable discussion and conclusions.
The following suggestions are provided to assist fine tuning the manuscript.
L3- Suggest: hormones for hormone
L70- Suggest: has also been instead of has been also
L84- delete the in: mechanisms of flower...
L91- meristem vs mersitems...were vs was...
L92- floral buds....flower buds elongated...
L96- petal fall for petals
L124, L139, L151, L291- Because these are headings it would be helpful to the reader to spell out DEGs.
L227- Suggest adding A, B, C to the legend in Fig. 9
L233- shown vs showed..
L234- delete were
L260- delete the
L296- delete the before flower..
L373- Suggest: Proteins were extracted from floral....
L377- delete for...
L380 - use with instead of by
L391- extinction for extinctions.
Author Response
Dear Editor and Reviewers,
First of all, thank you very much for the minor revisions of your letter and the reviewers’ advice and comments to improve the manuscript “An Integrative Analysis of Transcriptome, Proteome and Hormone Reveals Key Differentially Expressed Genes and Metabolic Pathways Involved in Flower Development in Loquat” (Manuscript ID: ijms-848521-Minor Revisions). These comments are all valuable and very helpful for improving our paper, as well as the important guiding significance to our study. We have studied the comments carefully, and revised our manuscript according to your comments. We hope our revised manuscript has met with the approval. The edited comments were highlighted using red text in the revised manuscript. The main corrections in the manuscript and the responds to your comments are as following.
Now, we give a point-by-point response to the comments:
Response to Reviewer 2
Point 1: This is a high quality manuscript based upon a comprehensive, well designed study that presents many lines of converging evidence. The work provides considerable data that enlightens the molecular/genetic knowledge base supporting floral initiation and development in Loquat. It very nicely links gene action to hormonal changes and morphological development in early flowering. The paper is well-written with reasonable discussion and conclusions.
Response 1: Thanks very much for your positive affirmation and evaluation for our study. We have revised manuscript as your good suggestions and give a point-by-point response.
Point 2: The following suggestions are provided to assist fine tuning the manuscript. L3- Suggest: hormones for hormone
Response 2: Thanks for this very good suggestion. We have changed “hormone” to “hormones” in our revised manuscript.
Point 3: L70- Suggest: has also been instead of has been also
Response 3: According to the advice, we have changed “has been also” to “has also been”.
Point 4: L84- delete the in: mechanisms of flower...
Response 4: According to the advice, we have deleted “the” in “mechanisms of flower” in the revised manuscript.
Point 5: L91- meristem vs meristems...were vs was...
Response 5: According to the advice, we have changed “meristems” to “meristem” and “was” to “were” in our revised manuscript.
Point 6: L92- floral buds....flower buds elongated...
Response 6: Thanks for this very good suggestion. In our revised manuscript, we have change “floral bud” to “floral buds” and “flower bud elongated” to “flower buds elongated”.
Point 7: L96- petal fall for petals
Response 7: According to the advice, we have changed “petals fall” to “petal fall” in the revised manuscript.
Point 8: L124, L139, L151, L291- Because these are headings it would be helpful to the reader to spell out DEGs.
Response 8: Thanks for this very good suggestion. According to the advice, we have spell out DEGs “differentially expressed genes” in L124, L139, L151, L291 to be helpful to the reader.
Point 9: L227- Suggest adding A, B, C to the legend in Fig. 9
Response 9: Thanks for this very good suggestion. In our revised manuscript, we have added the correlation analysis for DAPs and DEGs of FBE vs FBD, FA vs FBE and FA vs FBD in Fig. 8 and the legend of A, B, C, D, E and F in Fig. 8 and the legend of A, B, C in Fig. 9.
Point 10: L233- shown vs showed.
Response 10: According to the advice, we have changed “shown” to “showed” in our revised manuscript.
Point 11: L234- delete were
Response 11: In our revised manuscript, we have deleted “were”.
Point 12: L260- delete the
Response 12: Thanks for this very good suggestion. In our revised manuscript, we have deleted “the”.
Point 13: L296- delete the before flower…
Response 13: According to the advice, we have deleted “the” in the revised manuscript.
Point 14: L373- Suggest: Proteins were extracted from floral...
Response 14: Thanks for this very good suggestion. In our revised manuscript, we have change to “Proteins were extracted from floral...”.
Point 15: L377- delete for...
Response 15: According to the advice, we have deleted “for three times ”in the revised manuscript.
Point 16: L380 – use with instead of by
Response 16: Thanks for this very good suggestion. We have changed “by” to “use with” in our revised manuscript.
Point 17: L391- extinction for extinctions.
Response 17: We have revised the inaccurate English expression and changed “extinctions” to “extinction” in our revised manuscript.
We try our best to improve the manuscript and look forward to your positive response. Thank you very much for your consideration.
Danlong Jing
Postal address: College of Horticulture and Landscape Architecture, Southwest University, Beibei, Chongqing 400715, China.
Email: jingdanlong110@126.com
Tel.: +86 023 68250383

This manuscript is a resubmission of an earlier submission. The following is a list of the peer review reports and author responses from that submission.
Round 1
Reviewer 1 Report
This manuscript describes an analysis of the floral transcriptomes of loquat at three developmental time-points. The manuscript is largely descriptive, but that is the mature of such work, and it is novel and as far as I can assess has been performed carefully and rigorously. qPCR has also been used to validate the differential expression results. Overall I judge that this work has considerable merit particularly as it focuses on a plant species of some economic importance.
I have two main overall comments:
- Other recently published (one in this journal) loquat transcriptome manuscripts should be mentioned and referenced - they are involved in fruit development rather than flower development, but are part of the sequence resources that this manuscript adds to, with a quick Google search I found these two, the first seems to highlight sets of DEGs……
10.3390/ijms17111837
10.3389/fpls.2016.01924
- The manuscript has been carefully prepared, but there are numerous grammatical problems (some sections are worse than others).
I have made a (rather long) list of things that need to be addressed below:
Line 24: “Gibberellin A (GA)” – is this correct? The abbreviation GA is more usually used for “gibberellic acid”.
Line 56/57 “Considerable progresses of flowering mechanism in model plants have greatly facilitated our understanding on the molecular networks of flower development [9, 19, 20].”
Change to: “Considerable progress in elucidating the flowering mechanism in model plants has greatly improved our understanding of the molecular networks of flower development [9, 19, 20].”
Line 60: “Transcriptome is an extremely effective method for obtaining differentially ……..”
Change to: “Transcriptomics are an extremely effective method for identifying differentially ……..”
Line 80: “To obtain the morphological characteristics, we observed morphological changes during flower development process.”
Change to: “To characterize flower development, we assessed morphological changes during the development process.”
Line 82: “Nine morphologically distinct stages were defined from the vegetative apex to the petals fall.”
Change to: ““Nine morphologically distinct stages were defined.”
Line 95 “To further gain the transcriptomic changes during the flower development, cDNA libraries were prepared independently from flower bud differentiation (FBD, stage 2), floral bud elongation (FBE, stage 5) and floral anthesis (FA, stage 8) based on morphological division.”
Change to: ““To gain insight into the transcriptomic changes during flower development, cDNA libraries were prepared independently from flower bud differentiation (FBD, stage 2), floral bud elongation (FBE, stage 5) and floral anthesis (FA, stage 8).”
Line 102: “matched to the Nr database by the BLAST analysis…”
Change to: “matched to the Nr database by BLAST analysis…”
Line 149: “In SA-signaling pathway, SABP2 were identified”
Change to “In SA-signaling pathway, SABP2 was identified”
Line 173: “genes regulating the flower development”
Chage to: “genes regulating flower development”
Line 175: “further analyzed the five most number of DEGs in TF families.”
I am not certain what this is supposed to mean is it “further analyzed the five TF families most highly represented in the DEGs.” ?
Line 202: “The results reveal that key DEGs of floral integrator, floral meristem identity, floral organ identity genes and hormone signal transduction pathways associated with flower development (Figure 9).”
This is a confusing sentence – suggestion:
“The results reveal that DEGS include floral integrator, floral meristem identity, floral organ identity genes, and genes involved in hormone signal transduction pathways associated with flower development (Figure 9).”
Line 223 “In our work, expression level of floral organ identity genes…”
Change to:
“In our work, expression levels of floral organ identity genes…”
Line 225: “Previously, AG, AP3, PI and AGL orthologs are mainly transcribed in flower organs and play crucial roles in floral organ identity specification. For example, expression analysis of AG orthologs were strongly detected in the reproductive organs including stamens and carpels in different clades of angiosperms, ….”
Change to: “Previously, AG, AP3, PI and AGL orthologs have been reported to be mainly transcribed in floral organs and play crucial roles in organ identity specification. For example, expression of AG orthologs was strongly detected in reproductive organs including stamens and carpels in different clades of angiosperms, ….”
Line 237: “Plant hormones participate in all stages of the flower development. In our study, the DEGs, which involved in plant hormone signal transduction, were significantly enriched in the auxin, GA, cytokinin, enthylene, ABA, JA and SA signaling pathways during flower development in loquat. Accordingly, concentrations of GA3 and ZT were high before the FA stage, but ABA concentration remained high at the FA stage. This indicated that the flower development is involved in the effects of endogenous hormones in loquat.”
Suggestion based on what I think you mean: “Plant hormones participate in all stages of flower development. In our study, DEGs involved in auxin, GA, cytokinin, ethylene, ABA, JA and SA signaling pathways, were significantly enriched during flower development in loquat. Consistent with these results, concentrations of GA3 and ZT were high before the FA stage, but ABA concentration remained high at the FA stage.”
Line 241: “This indicated that the flower development is involved in the effects of endogenous hormones in loquat.”
- I really cannot make sense of this statement….
Line 247 “In Arabidopsis, GAs were regulated floral integrator genes SOC1 and FT and floral meristem identity gene LFY to promote 247 floral induction [51-53].”
- Again I cannot make sense of this statement….
Line 250: “demonstrated that flower organs development”
Change to: “demonstrated that floral organ development”
Line 267: “The increase in ABA concentration in the later stage of loquat flower development may be involved in the effect of ABA on senescence.”
Change to: “The increase in ABA concentration in the later stage of loquat flower development may be related to the involvement of ABA in petal senescence.”
Line 281: “Extration kit” change to “Extraction kit”
Line 317: “0.3 g each sample was…” change to “).3 g of each sample was…”
Line 341: “Table S8. The five most number of DEGs in TF families.”
Change to: “Table S8. The five TF families most highly represented in the DEGs.”
Author Response
Dear Reviewer:
Thank you very much for your comments concerning our manuscript entitled “Transcriptomic and Hormone Analyses Reveal Key Differentially Expressed Genes and Metabolic Pathways Involved in Flower Development in Loquat” (ID: ijms-809959). These comments are all valuable and very helpful for improving our paper, as well as the important guiding significance to our researches. We have studied the comments carefully, and revised our manuscript according to your comments. We hope our revised manuscript has met with the approval. The edited comments were highlighted using red text in the revised manuscript. The main corrections in the manuscript and the responds to your comments are as following.
Now, we give a point-by-point response to the comments:
Point 1: This manuscript describes an analysis of the floral transcriptomes of loquat at three developmental time-points. The manuscript is largely descriptive, but that is the mature of such work, and it is novel and as far as I can assess has been performed carefully and rigorously. qPCR has also been used to validate the differential expression results. Overall I judge that this work has considerable merit particularly as it focuses on a plant species of some economic importance.
Response 1: Thanks very much for your positive affirmation and evaluation for our study. We have revised manuscript as your good suggestions and give a point-by-point response.
Point 2: I have two main overall comments:
Other recently published (one in this journal) loquat transcriptome manuscripts should be mentioned and referenced - they are involved in fruit development rather than flower development, but are part of the sequence resources that this manuscript adds to, with a quick Google search I found these two, the first seems to highlight sets of DEGs……
10.3390/ijms17111837
10.3389/fpls.2016.01924
Response 1: Thanks for this very good suggestion. We have added the two references, which you have suggested, including:
[24]. Song H, Zhao X, Hu W, Wang X, Shen T, Yang L. Comparative Transcriptional Analysis of Loquat Fruit Identifies Major Signal Networks Involved in Fruit Development and Ripening Process. Int J Mol Sci. 2016, 17(11), 1837.
[25] Jiang S, Luo J, Xu F, Zhang X. Transcriptome Analysis Reveals Candidate Genes Involved in Gibberellin-Induced Fruit Setting in Triploid Loquat (Eriobotrya japonica). Front Plant Sci. 2016, 7, 1924.”.
Meanwhile, according to the advices, we have added the comment in the Introduction “At present, transcriptomic analyses have been mainly reported in fruit development of loquat, and provide sequence resources that are involved in fruit setting, development and ripening process [24, 25].” and in the Conclusions “Compared with transcriptome analysis of loquat fruit development in recent time [24, 25], our results provide key DEGs…”.
Point 3: The manuscript has been carefully prepared, but there are numerous grammatical problems (some sections are worse than others).
I have made a (rather long) list of things that need to be addressed below: Line 24: “Gibberellin A (GA)” – is this correct? The abbreviation GA is more usually used for “gibberellic acid”.
Response 3: We have revised the inaccurate English expression and changed “gibberellin A (GA)” to “gibberellic acid (GA)” in our revised manuscript.
Point 4: Line 56/57 “Considerable progresses of flowering mechanism in model plants have greatly facilitated our understanding on the molecular networks of flower development [9, 19, 20].”
Change to: “Considerable progress in elucidating the flowering mechanism in model plants has greatly improved our understanding of the molecular networks of flower development [9, 19, 20].”
Response 4: It is a very good suggestion to revise unprecise English usage. According to the reviewer’s advice, we have changed to “Considerable progress in elucidating the flowering mechanism in model plants has greatly improved our understanding of the molecular networks of flower development [9, 19, 20].” in our revised manuscript.
Point 5: Line 60: “Transcriptome is an extremely effective method for obtaining differentially ……..”
Change to: “Transcriptomics are an extremely effective method for identifying differentially ……..”
Response 5: We have changed to “Transcriptomics is an extremely effective method for identifying differentially …” in our revised manuscript.
Point 6: Line 80: “To obtain the morphological characteristics, we observed morphological changes during flower development process.”
Change to: “To characterize flower development, we assessed morphological changes during the development process.”
Response 6: Thanks for this very good suggestion. In our revised manuscript, we have change to “To characterize flower development, we assessed morphological changes during the development process.”
Point 7: Line 82: “Nine morphologically distinct stages were defined from the vegetative apex to the petals fall.”
Change to: “Nine morphologically distinct stages were defined.”
Response 7: Thanks for this very good suggestion. According to the advice, we have changed to “Nine morphologically distinct stages were defined.”
Point 8: Line 95 “To further gain the transcriptomic changes during the flower development, cDNA libraries were prepared independently from flower bud differentiation (FBD, stage 2), floral bud elongation (FBE, stage 5) and floral anthesis (FA, stage 8) based on morphological division.”
Change to: ““To gain insight into the transcriptomic changes during flower development, cDNA libraries were prepared independently from flower bud differentiation (FBD, stage 2), floral bud elongation (FBE, stage 5) and floral anthesis (FA, stage 8).”
Response 8: In our revised manuscript, we have changed to “To gain insight into the transcriptomic changes during flower development, cDNA libraries were prepared independently from flower bud differentiation (FBD, stage 2), floral bud elongation (FBE, stage 5) and floral anthesis (FA, stage 8).”
Point 9: Line 102: “matched to the Nr database by the BLAST analysis…”
Change to: “matched to the Nr database by BLAST analysis…”
Response 9: Thank you for your guidance. We have changed to “matched to the Nr database by BLAST analysis…” in the revised manuscript.
Point 10: Line 149: “In SA-signaling pathway, SABP2 were identified”
Change to “In SA-signaling pathway, SABP2 was identified”
Response 10: Thanks for this very good suggestion. We have revised manuscript as this advice, and changed to “In SA-signaling pathway, SABP2 was identified”.
Point 11: Line 173: “genes regulating the flower development”
Change to: “genes regulating flower development”
Response 11: Thanks to this very good suggestion. According to the advice, we have changed to “genes regulating flower development”.
Point 12: Line 175: “further analyzed the five most number of DEGs in TF families.”
I am not certain what this is supposed to mean is it “further analyzed the five TF families most highly represented in the DEGs.” ?
Response 12: According to the advice, we have changed to “further analyzed the five TF families most highly represented in the DEGs.” in the revised manuscript.
Point 13: Line 202: “The results reveal that key DEGs of floral integrator, floral meristem identity, floral organ identity genes and hormone signal transduction pathways associated with flower development (Figure 9).”
This is a confusing sentence – suggestion: “The results reveal that DEGs include floral integrator, floral meristem identity, floral organ identity genes, and genes involved in hormone signal transduction pathways associated with flower development (Figure 9).”
Response 13: Thanks for this very good suggestion. We have changed to “The results reveal that DEGs include floral integrator, floral meristem identity, floral organ identity genes, and genes involved in hormone signal transduction pathways associated with flower development (Figure 9).” in the revised manuscript.
Point 14: Line 223 “In our work, expression level of floral organ identity genes…”
Change to: “In our work, expression levels of floral organ identity genes…”
Response 14: In our revised manuscript, we have changed to “In our work, expression levels of floral organ identity genes…”.
Point 15: Line 225: “Previously, AG, AP3, PI and AGL orthologs are mainly transcribed in flower organs and play crucial roles in floral organ identity specification. For example, expression analysis of AG orthologs were strongly detected in the reproductive organs including stamens and carpels in different clades of angiosperms, ….”
Change to: “Previously, AG, AP3, PI and AGL orthologs have been reported to be mainly transcribed in floral organs and play crucial roles in organ identity specification. For example, expression of AG orthologs was strongly detected in reproductive organs including stamens and carpels in different clades of angiosperms, ….”
Response 15: Thanks for the very good suggestion. According to the advice, we have changed to “Previously, AG, AP3, PI and AGL orthologs have been reported to be mainly transcribed in floral organs and play crucial roles in organ identity specification. For example, expression of AG orthologs was strongly detected in reproductive organs including stamens and carpels in different clades of angiosperms, ….”
Point 16: Line 237: “Plant hormones participate in all stages of the flower development. In our study, the DEGs, which involved in plant hormone signal transduction, were significantly enriched in the auxin, GA, cytokinin, enthylene, ABA, JA and SA signaling pathways during flower development in loquat. Accordingly, concentrations of GA3 and ZT were high before the FA stage, but ABA concentration remained high at the FA stage. This indicated that the flower development is involved in the effects of endogenous hormones in loquat.”
Suggestion based on what I think you mean: “Plant hormones participate in all stages of flower development. In our study, DEGs involved in auxin, GA, cytokinin, ethylene, ABA, JA and SA signaling pathways, were significantly enriched during flower development in loquat. Consistent with these results, concentrations of GA3 and ZT were high before the FA stage, but ABA concentration remained high at the FA stage.”
Response 16: Thanks for this very good suggestion. According to the advice, we have changed to “Plant hormones participate in all stages of flower development. In our study, DEGs involved in auxin, GA, cytokinin, ethylene, ABA, JA and SA signaling pathways, were significantly enriched during flower development in loquat. Consistent with these results, concentrations of GA3 and ZT were high before the FA stage, but ABA concentration remained high at the FA stage.”.
Point 17: Line 241: “This indicated that the flower development is involved in the effects of endogenous hormones in loquat.” I really cannot make sense of this statement….
Response 17: In the revised manuscript, we have changed to “This indicated that the flower development is related to the involvement of endogenous hormones in loquat.”
Point 18: Line 247 “In Arabidopsis, GAs were regulated floral integrator genes SOC1 and FT and floral meristem identity gene LFY to promote 247 floral induction [51-53].”
Again I cannot make sense of this statement….
Response 18: We have changed to “In Arabidopsis, GAs control floral induction through regulating floral integrator genes SOC1 and FT and floral meristem identity gene LFY” in our revised manuscript.
Point 19: Line 250: “demonstrated that flower organs development”
Change to: “demonstrated that floral organ development”
Response 19: Thanks for the very good suggestion. We have changed to “demonstrated that floral organ development”.
Point 20: Line 267: “The increase in ABA concentration in the later stage of loquat flower development may be involved in the effect of ABA on senescence.”
Change to: “The increase in ABA concentration in the later stage of loquat flower development may be related to the involvement of ABA in petal senescence.”
Response 20: We have revised the inaccurate English expression, and changed to “The increase in ABA concentration in the later stage of loquat flower development may be related to the involvement of ABA in petal senescence.”
Point 21: Line 281: “Extration kit” change to “Extraction kit”
Response 21: We have revised these typo errors, and have changed to “Extraction kit” in the revised manuscript.
Point 22: Line 317: “0.3 g each sample was…” change to “0.3 g of each sample was…”
Response 22: In the revised manuscript, we have changed to “0.3 g of each sample was…”.
Point 23: Line 341: “Table S8. The five most number of DEGs in TF families.”
Change to: “Table S8. The five TF families most highly represented in the DEGs.”
Response 23: Thank you for this very good suggestion. According to the advice, we have changed to “Table S8. The five TF families most highly represented in the DEGs.”
We try our best to improve the manuscript and look forward to your positive response. Thank you very much for your consideration.

Reviewer 2 Report
I found this article extremely difficult to read as well as redundant with respect to the current literature already available on the topic. Extensive editing of grammar and style is needed before the manuscript can be properly reviewed.
I am not overly impressed by the content of the manuscript as well, as it replicates many of the recurring experiments which are currently populating the submission platform of IJMS. This manuscript is indeed another example of RNAseq ran in three stages of flower development that reveals a high occurrence of genes of the class of hormone biosynthesis. Not to mention that hormones have not even been measured. In addition, the manuscript does add to the full understanding of the developmental process of flowers of woody plants and loquat in particular.
Additional relevant comments and recurring inaccuracies are reported below:
- I do not understand the relevance and the meaning of the “validation of expression…” experiment (lines 178 to 183). This part can be omitted.
- A proper scale should be provided in Figures 4, 5, and 6.
- The title of the manuscript is overrated. There is no mention or measurement of flower metabolites and hormones; therefore, the “metabolic pathway” part of the title should be omitted.
- The word “regulator” is often used with the meaning of “gene involved in the biosynthesis of.” These are two different concepts
- Many adverbs are improperly used.
- All genes must be written in full and an abbreviation provided.
- “expression dynamics of TF…” is inaccurate as this experiment only provides steady-state levels of transcripts.
- A definition of integrated genes must be provided.
Author Response
Dear Reviewer:
Thank you very much for your comments concerning our manuscript entitled “Transcriptomic and Hormone Analyses Reveal Key Differentially Expressed Genes and Metabolic Pathways Involved in Flower Development in Loquat” (ID: ijms-809959). These comments are all valuable and very helpful for improving our paper, as well as the important guiding significance to our researches. We have studied the comments carefully, and revised our manuscript according to your comments. We hope our revised manuscript has met with the approval. The edited comments were highlighted using red text in the revised manuscript. The main corrections in the manuscript and the responds to your comments are as following.
Now, we give a point-by-point response to Reviewer Comments:
Point 1: I found this article extremely difficult to read as well as redundant with respect to the current literature already available on the topic. Extensive editing of grammar and style is needed before the manuscript can be properly reviewed.
Response 1: It is a very good suggestion to revise unprecise English usage. The English in this document was edited for proper English language, grammar, punctuation, spelling, and overall style. These editing comments are as following:
- We have changed Line 56/57 “Considerable progresses of flowering mechanism in model plants have greatly facilitated our understanding on the molecular networks of flower development [9, 19, 20].” to “Considerable progress in elucidating the flowering mechanism in model plants has greatly improved our understanding of the molecular networks of flower development [9, 19, 20].”
- We have changed Line 60: “Transcriptome is an extremely effective method for obtaining differentially ……..” to “Transcriptomics is an extremely effective method for identifying differentially ……..”
- We have changed Line 80: “To obtain the morphological characteristics, we observed morphological changes during flower development process.” to “To characterize flower development, we assessed morphological changes during the development process.”
- We have changed Line 82: “Nine morphologically distinct stages were defined from the vegetative apex to the petals fall.” to “Nine morphologically distinct stages were defined.”
- We have changed Line 95 “To further gain the transcriptomic changes during the flower development, cDNA libraries were prepared independently from flower bud differentiation (FBD, stage 2), floral bud elongation (FBE, stage 5) and floral anthesis (FA, stage 8) based on morphological division.” to “To gain insight into the transcriptomic changes during flower development, cDNA libraries were prepared independently from flower bud differentiation (FBD, stage 2), floral bud elongation (FBE, stage 5) and floral anthesis (FA, stage 8).”
- We have changed Line 102: “matched to the Nr database by the BLAST analysis…” to “matched to the Nr database by BLAST analysis…”
- We have changed Line 149: “In SA-signaling pathway, SABP2 were identified” to “In SA-signaling pathway, SABP2 was identified”.
- We have changed Line 173: “genes regulating the flower development” to “genes regulating flower development”.
- We have changed Line 175: “further analyzed the five most number of DEGs in TF families.” to “further analyzed the five TF families most highly represented in the DEGs.”.
- We have changed Line 202: “The results reveal that key DEGs of floral integrator, floral meristem identity, floral organ identity genes and hormone signal transduction pathways associated with flower development (Figure 9).” to “The results reveal that DEGS include floral integrator, floral meristem identity, floral organ identity genes, and genes involved in hormone signal transduction pathways associated with flower development (Figure 9).”
- We have changed Line 223 “In our work, expression level of floral organ identity genes…” to “In our work, expression levels of floral organ identity genes…”.
- We have changed Line 225: “Previously, AG, AP3, PI and AGL orthologs are mainly transcribed in flower organs and play crucial roles in floral organ identity specification. For example, expression analysis of AG orthologs were strongly detected in the reproductive organs including stamens and carpels in different clades of angiosperms, ….” to “Previously, AG, AP3, PI and AGL orthologs have been reported to be mainly transcribed in floral organs and play crucial roles in organ identity specification. For example, expression of AG orthologs was strongly detected in reproductive organs including stamens and carpels in different clades of angiosperms, ….”
- We have changed Line 237: “… In our study, the DEGs, which involved in plant hormone signal transduction, were significantly enriched in the auxin, GA, cytokinin, enthylene, ABA, JA and SA signaling pathways during flower development in loquat. Accordingly, concentrations of GA3 and ZT were high before the FA stage, but ABA concentration remained high at the FA stage. This indicated that the flower development is involved in the effects of endogenous hormones in loquat.” to “… In our study, DEGs involved in auxin, GA, cytokinin, ethylene, ABA, JA and SA signaling pathways, were significantly enriched during flower development in loquat. Consistent with these results, concentrations of GA3 and ZT were high before the FA stage, but ABA concentration remained high at the FA stage.”
- We have changed Line 241: “This indicated that the flower development is involved in the effects of endogenous hormones in loquat.” to “This indicated that the flower development is related to the involvement of endogenous hormones in loquat.”
- We have changed Line 247 “In Arabidopsis, GAs were regulated floral integrator genes SOC1 and FT and floral meristem identity gene LFY to promote 247 floral induction [51-53].” to “In Arabidopsis, GAs control floral induction through regulating floral integrator genes SOC1 and FT and floral meristem identity gene LFY” in our revised manuscript.”.
- We have changed Line 250: “demonstrated that flower organs development” to “demonstrated that floral organ development”.
- We have changed Line 267: “The increase in ABA concentration in the later stage of loquat flower development may be involved in the effect of ABA on senescence.” to “The increase in ABA concentration in the later stage of loquat flower development may be related to the involvement of ABA in petal senescence.”.
- We have changed Line 281: “Extration kit” to “Extraction kit”.
- We have changed Line 317: “0.3 g each sample was…” to “0.3 g of each sample was…”.
- We have changed Line 341: “Table S8. The five most number of DEGs in TF families.” to “Table S8. The five TF families most highly represented in the DEGs.”.
Point 2: I am not overly impressed by the content of the manuscript as well, as it replicates many of the recurring experiments which are currently populating the submission platform of IJMS. This manuscript is indeed another example of RNAseq ran in three stages of flower development that reveals a high occurrence of genes of the class of hormone biosynthesis. Not to mention that hormones have not even been measured. In addition, the manuscript does add to the full understanding of the developmental process of flowers of woody plants and loquat in particular.
Response 2: Thanks very much for your positive affirmation and evaluation for our study. We have revised manuscript as your good suggestions and give a point-by-point response.
Point 3: Additional relevant comments and recurring inaccuracies are reported below:
I do not understand the relevance and the meaning of the “validation of expression…” experiment (lines 178 to 183). This part can be omitted.
Response 3: This part is to validate the expression levels of several key DEGs obtained by RNAseq using qRT-PCR method. Therefore, we have added the comments “The same change trends of these DEGs were showed between qRT-PCR and FPKM values (Figure 7), suggesting the expression trends of most unigenes were corresponded well between the two methods.”
Point 4: A proper scale should be provided in Figures 4, 5, and 6.
Response 4: Thanks for this very good suggestion. According to the advice, we have added the scale, which have been provided in Figures 4, 5, and 6 in our revised manuscript.
Point 5: The title of the manuscript is overrated. There is no mention or measurement of flower metabolites and hormones; therefore, the “metabolic pathway” part of the title should be omitted.
Response 5: In our revised manuscript, we have omitted the “metabolic pathway” part of the title, and changed to “Transcriptomic and Hormone Analyses Reveal Key Differentially Expressed Genes and Hormonal Control Involved in Flower Development in Loquat”.
Point 6: The word “regulator” is often used with the meaning of “gene involved in the biosynthesis of.” These are two different concepts
Many adverbs are improperly used.
Response 6: In our revised manuscript, we have revised the inaccurate English expression and changed “regulator” to “gene”.
Point 7: All genes must be written in full and an abbreviation provided.
Response 7: Thanks for this very good suggestion. According to the reviewer’s advices, we have added the full and an abbreviation in our revised manuscript and the Abbreviations part.
Point 8: “expression dynamics of TF…” is inaccurate as this experiment only provides steady-state levels of transcripts.
Response 8: It is a very good suggestion to revise inaccurate English usage. According to the reviewer’s advice, we have replaced “expression dynamics of TF…” to “expression levels of TF…”.
Point 9: A definition of integrated genes must be provided.
Response 9: In our revised manuscript, we have provided a definition of integrated genes and changed to “All of these pathways converge to activate a small number of floral integrator genes, which control floral development by merging signals from various pathways.” in the introduction.
We try our best to improve the manuscript and look forward to your positive response. Thank you very much for your consideration.
Reviewer 3 Report
Review Comments
In this manuscript, the authors used the transcriptomic and hormone analyses to investigate the key candidate genes and regulatory pathways during flower development in loquat (Eriobotrya japonica). Comparative transcriptome analysis showed that the differentially expressed genes (DEGs) were mainly enriched in metabolic pathways of hormone signal transduction, and starch and sucrose metabolism. Key floral integrator genes FT and SOC1 and floral meristem identity genes SPLs, LFY, AP1 and AP2 were significantly up-regulated at flower bud differentiation (FBD) stage. However, key floral organ identity genes AG, AP3 and PI were significantly up-regulated at the stages of floral bud elongation (FBE) and floral anthesis (FA). Moreover, transcription factor genes such as bHLH, NAC, MYB_related, ERF and C2H2, were also significantly differentially expressed. These results provide abundant sequence resources for clarifying the underlying mechanisms of the flower development in loquat.
In fact, the manuscript is well designed and written and shows sort of novelty that will help in enhancing flowers development. However, the following revisions should be taken into consideration;
Please revise the genes names and make them italic throughout the whole manuscript.
Also, please revise English and correct the mistakes in the grammar of this section.
- Methods:
Methods section requires more details on how to prepare the samples to carry out these measurements.
Statistical analysis should be mentioned in details, including the programs and versions used.
- Results
The data are well represented. Figures are well explained. However, some figures such as Figure 3 should be replaced with a high resolution one and more explained.
- Discussion
The discussion section is well written. But it's better to add a few sentences revealing the link between the current data with the previous findings.
The conclusion section should be improved to highlight the significant data and findings arose in this study.
Author Response
Dear Reviewer:
Thank you very much for your comments concerning our manuscript entitled “Transcriptomic and Hormone Analyses Reveal Key Differentially Expressed Genes and Metabolic Pathways Involved in Flower Development in Loquat” (ID: ijms-809959). These comments are all valuable and very helpful for improving our paper, as well as the important guiding significance to our researches. We have studied the comments carefully, and revised our manuscript according to your comments. We hope our revised manuscript has met with the approval. The edited comments were highlighted using red text in the revised manuscript. The main corrections in the manuscript and the responds to your comments are as following.
Now, we give a point-by-point response to Reviewer Comments:
Point 1: In this manuscript, the authors used the transcriptomic and hormone analyses to investigate the key candidate genes and regulatory pathways during flower development in loquat (Eriobotrya japonica). Comparative transcriptome analysis showed that the differentially expressed genes (DEGs) were mainly enriched in metabolic pathways of hormone signal transduction, and starch and sucrose metabolism. Key floral integrator genes FT and SOC1 and floral meristem identity genes SPLs, LFY, AP1 and AP2 were significantly up-regulated at flower bud differentiation (FBD) stage. However, key floral organ identity genes AG, AP3 and PI were significantly up-regulated at the stages of floral bud elongation (FBE) and floral anthesis (FA). Moreover, transcription factor genes such as bHLH, NAC, MYB_related, ERF and C2H2, were also significantly differentially expressed. These results provide abundant sequence resources for clarifying the underlying mechanisms of the flower development in loquat.
In fact, the manuscript is well designed and written and shows sort of novelty that will help in enhancing flowers development. However, the following revisions should be taken into consideration;
Response 1: Thanks very much for your positive affirmation and evaluation for our study. We have revised manuscript as your good suggestions and give a point-by-point response.
Point 2: Please revise the genes names and make them italic throughout the whole manuscript.
Response 2: According to this very good suggestion, we have checked and revised the genes names, and made them italic in the whole revised manuscript.
Point 3: Also, please revise English and correct the mistakes in the grammar of this section.
Response 3: We have revise unprecise English usage and numerous grammatical problems. These editing comments are as following:
- We have changed Line 56/57 “Considerable progresses of flowering mechanism in model plants have greatly facilitated our understanding on the molecular networks of flower development [9, 19, 20].” to “Considerable progress in elucidating the flowering mechanism in model plants has greatly improved our understanding of the molecular networks of flower development [9, 19, 20].”
- We have changed Line 60: “Transcriptome is an extremely effective method for obtaining differentially ……..” to “Transcriptomics is an extremely effective method for identifying differentially ……..”
- We have changed Line 80: “To obtain the morphological characteristics, we observed morphological changes during flower development process.” to “To characterize flower development, we assessed morphological changes during the development process.”
- We have changed Line 82: “Nine morphologically distinct stages were defined from the vegetative apex to the petals fall.” to “Nine morphologically distinct stages were defined.”
- We have changed Line 95 “To further gain the transcriptomic changes during the flower development, cDNA libraries were prepared independently from flower bud differentiation (FBD, stage 2), floral bud elongation (FBE, stage 5) and floral anthesis (FA, stage 8) based on morphological division.” to “To gain insight into the transcriptomic changes during flower development, cDNA libraries were prepared independently from flower bud differentiation (FBD, stage 2), floral bud elongation (FBE, stage 5) and floral anthesis (FA, stage 8).”
- We have changed Line 102: “matched to the Nr database by the BLAST analysis…” to “matched to the Nr database by BLAST analysis…”
- We have changed Line 149: “In SA-signaling pathway, SABP2 were identified” to “In SA-signaling pathway, SABP2 was identified”.
- We have changed Line 173: “genes regulating the flower development” to “genes regulating flower development”.
- We have changed Line 175: “further analyzed the five most number of DEGs in TF families.” to “further analyzed the five TF families most highly represented in the DEGs.”.
- We have changed Line 202: “The results reveal that key DEGs of floral integrator, floral meristem identity, floral organ identity genes and hormone signal transduction pathways associated with flower development (Figure 9).” to “The results reveal that DEGS include floral integrator, floral meristem identity, floral organ identity genes, and genes involved in hormone signal transduction pathways associated with flower development (Figure 9).”
- We have changed Line 223 “In our work, expression level of floral organ identity genes…” to “In our work, expression levels of floral organ identity genes…”.
- We have changed Line 225: “Previously, AG, AP3, PI and AGL orthologs are mainly transcribed in flower organs and play crucial roles in floral organ identity specification. For example, expression analysis of AG orthologs were strongly detected in the reproductive organs including stamens and carpels in different clades of angiosperms, ….” to “Previously, AG, AP3, PI and AGL orthologs have been reported to be mainly transcribed in floral organs and play crucial roles in organ identity specification. For example, expression of AG orthologs was strongly detected in reproductive organs including stamens and carpels in different clades of angiosperms, ….”
- We have changed Line 237: “… In our study, the DEGs, which involved in plant hormone signal transduction, were significantly enriched in the auxin, GA, cytokinin, enthylene, ABA, JA and SA signaling pathways during flower development in loquat. Accordingly, concentrations of GA3 and ZT were high before the FA stage, but ABA concentration remained high at the FA stage. This indicated that the flower development is involved in the effects of endogenous hormones in loquat.” to “… In our study, DEGs involved in auxin, GA, cytokinin, ethylene, ABA, JA and SA signaling pathways, were significantly enriched during flower development in loquat. Consistent with these results, concentrations of GA3 and ZT were high before the FA stage, but ABA concentration remained high at the FA stage.”
- We have changed Line 241: “This indicated that the flower development is involved in the effects of endogenous hormones in loquat.” to “This indicated that the flower development is related to the involvement of endogenous hormones in loquat.”
- We have changed Line 247 “In Arabidopsis, GAs were regulated floral integrator genes SOC1 and FT and floral meristem identity gene LFY to promote 247 floral induction [51-53].” to “In Arabidopsis, GAs control floral induction through regulating floral integrator genes SOC1 and FT and floral meristem identity gene LFY” in our revised manuscript.”.
- We have changed Line 250: “demonstrated that flower organs development” to “demonstrated that floral organ development”.
- We have changed Line 267: “The increase in ABA concentration in the later stage of loquat flower development may be involved in the effect of ABA on senescence.” to “The increase in ABA concentration in the later stage of loquat flower development may be related to the involvement of ABA in petal senescence.”.
- We have changed Line 281: “Extration kit” to “Extraction kit”.
- We have changed Line 317: “0.3 g each sample was…” to “0.3 g of each sample was…”.
- We have changed Line 341: “Table S8. The five most number of DEGs in TF families.” to “Table S8. The five TF families most highly represented in the DEGs.”.
Point 4: - Methods: Methods section requires more details on how to prepare the samples to carry out these measurements.
Response 4: In our revised manuscript, we have added more details in the Methods section.
“At each sampling point, the buds from about twenty panicles were collected and sampled.”, “Total RNA was extracted individually from FBD, FBE and FA, using the EASYspin Plant RNA Extraction kit (RN09, Aidlab, China), according to instructions from the manufacturer.”,
Point 5: Statistical analysis should be mentioned in details, including the programs and versions used.
Response 5: In our revised manuscript, we have added the details of statistical analysis “All data were analyzed using analysis of variance (ANOVA), and the differences were compared using PASW Statistics v18.0 software (SPSS Inc., Chicago, IL) and Duncan's multiple range test.”
Point 6: -Results: The data are well represented. Figures are well explained. However, some figures such as Figure 3 should be replaced with a high resolution one and more explained.
Response 6: In the revised Figure 3, we have increased font size and made sure the font is a high resolution and readable when printed. Meanwhile, in our revised manuscript, we have added the more comments “Figure 3. KEGG pathway enrichment analysis of DEGs. (A) Enrichment analysis of DEGs for FBE vs FBD. The pathway of plant hormone signal transduction was mainly enriched (black arrow). (B) Enrichment analysis of DEGs for FA vs FBE. The pathways of plant hormone signal transduction (black arrow) and starch and sucrose metabolism (green arrow) were mainly enriched.”
Point 7: -Discussion
The discussion section is well written. But it's better to add a few sentences revealing the link between the current data with the previous findings.
Response 7: According to this very good suggestion, we have add a few sentences in the discussion to revealing the link between the current data with the previous findings. For example, we have added the comments “Previous studies have shown that FT activates the floral integrator gene SOC1 and floral meristem gene AP1 to initiate the floral transition and flower bud differentiation in Arabidopsis [8, 26, 27]”, “Previously, AG, AP3, PI and AGL orthologs have been reported to be mainly transcribed in flower organs”, “In our study, DEGs involved in auxin, GA, cytokinin, enthylene, ABA, JA and SA signaling pathways, were significantly enriched during flower development in loquat. Consistent with these results,…”, “These similar results indicated that the RNA-Seq data and expression changes of the DEGs involved in floral transition are reliable in our study.”, “Consistent with these results, concentrations of GA3 and ZT were high before the FA stage, but ABA concentration remained high at the FA stage.” and “The difference of cytokinin biosynthesis and content between non-vernalization requiring loquat and vernalization requiring L. chinensis and B. napus”.
Point 8: The conclusion section should be improved to highlight the significant data and findings arose in this study.
Response 8: According to this very good suggestion, we have added the comments in conclusion section, and changed to “Different from previous transcriptome analysis of loquat fruit development [24, 25], our results provide key DEGs were mainly involved in…”, “The DEGs were significantly enriched in signaling pathways of auxin, GA, cytokinin, ethylene, ABA, jasmonic acid and salicylic acid. Consistent with these results, concentrations of GA3 and ZT were high before the FA stage, but ABA concentration remained high at the FA stage.” and “Taken together, these identified key genes and hormone signal transduction pathways increase our understanding…”.
We try our best to improve the manuscript and look forward to your positive response. Thank you very much for your consideration.

Round 2
Reviewer 3 Report
The manuscript has been greatly improved as per my recommended comments.